# Single versus combination treatment in tinnitus: an international, multicentre, parallel-arm, superiority, randomised controlled trial

Tinnitus is defined as the conscious awareness of a tonal or composite noise in the absence of a corresponding external acoustic source. This international multicentre, parallel-arm, superiority, randomised controlled trial investigated whether combination therapies are superior to single interventions in the treatment of chronic subjective tinnitus. Tinnitus patients were recruited from five clinical sites across the EU and randomly assigned using a web-based system, stratified by their hearing and distress level, to single or combination treatment of 12 weeks. Cognitive-behavioural therapy, hearing aids, app-based structured counselling, or app-based sound therapy were administered either alone or as a combination of two treatments resulting in ten treatment arms. App-based treatments were delivered without direct contact or guidance from clinicians. The primary outcome was the difference in the change from baseline to week 12 in the total score of the Tinnitus Handicap Inventory (THI) between single and combination treatments in the intention-to-treat population. All statistical analysis were performed blinded to treatment allocation. 674 patients of both sexes aged between 18 and 80 years were screened for eligibility. 461 participants (190 females) with chronic subjective tinnitus and at least mild tinnitus handicap were enroled, 230 of which were randomly assigned to single and 231 to combination treatment. Least-squares mean changes from baseline to week 12 were −11.7 for single treatment (95% confidence interval [CI], −14.4 to −9.0) and −14.9 for combination treatments (95% CI, −17.7 to −12.1), with a statistically significant group difference ($p = 0.034$). Cognitive-behavioural therapy and hearing aids alone had large effect sizes, which could not be further increased by combination treatment. No serious adverse events occurred. In this trial involving patients with chronic tinnitus, all treatment arms showed improvement in THI scores from baseline to week 12. Combination treatments showed a stronger clinical effect than single treatment, however, no clear synergistic effect was observed when combining treatments. Instead, we observed a compensatory effect, where a more effective treatment offsets the clinical effects of a less effective treatment. ClinicalTrials.gov Identifier: NCT04663828.

✉e-mail: stefan.schoisswohl@ukr.de

Tinnitus is defined as 'the conscious awareness of a tonal or composite noise for which there is no identifiable external acoustic source'[1], with an estimated prevalence of 14.4% (95% confidence interval [CI], 12.6–16.5) in the global population, with 2.3% (95% CI, 1.7–3.1) being severely affected[2]. Severe tinnitus is associated with emotional stress, cognitive dysfunction and/or autonomic arousal, leading to maladaptive behavioural changes and functional disability[1].

Numerous causes and risk factors for tinnitus have been identified[3], whereby peripheral and central mechanisms are involved in its emergence and maintenance, exemplified by pathological alterations in the ear, along the auditory pathway[4], as well as in non-auditory brain regions[5]. There is a broad spectrum of aetiologies, phenotypes and underlying pathophysiological mechanisms of tinnitus. Many adults with chronic tinnitus report having tried multiple tinnitus treatments before finding a treatment that reduces their tinnitus distress[6]. Despite the availability of guidelines[7,8], clear guidance on which treatment strategy is best for the individual patient is not yet available. A viable option for clinical management could be the combination of different treatment options to target various facets of this symptom simultaneously.

However, studies on the effectiveness of combining clinical interventions are scarce[9–11]. Prominent examples of combining different treatment types are represented by the combination of acoustic therapy with directive counselling as implemented in the Tinnitus Activities Treatment[12] or the Tinnitus Retraining Therapy[13].

The primary objective of the current trial was to investigate if combination treatments are more effective than single treatments for patients with chronic tinnitus. Four established treatment strategies were selected: cognitive-behavioural therapy (CBT), hearing aids (HA), app-based structured counselling (SC) and app-based sound therapy (ST)[14]. Participants were randomised either to a single treatment out of this set of treatments or to a combination of two treatments. Further, we attempt to overcome methodological weaknesses[15] of previous trials by investigating a large multinational sample of tinnitus patients, using harmonised patient selection and screening procedures, as well as standardised interventions and assessments.

## Results

Between Apr 16, 2021, and Sept 20, 2022, 674 persons with tinnitus were assessed for eligibility, of whom 461 (68.3%) fulfilled the inclusion criteria and consented to participate. After randomisation, 230 were allocated to single treatments and 231 were allocated to combination treatments (Fig. 1).

The initial planned sample size for the trial was 500 patients[16]. Since our study plan required a recruitment of an exact number of patients with specific tinnitus profiles (eligibility criteria and stratification proceedings), plus the trial was performed during the COVID-19 pandemic with country-specific hospital policies, recruitment and inclusion processes lasted longer than expected. Hence, we closed the trial in December 2022 with $N = 461$ included and treated patients, in order to keep to the schedule of our funding period (Granada: 89, Athens: 99, Leuven: 74, Regensburg: 100, Berlin: 99). A post hoc power computation indicates that with a two-tailed alpha level of less than 5%, the available sample size of $N = 461$ provides our trial with 79.5% power to detect an effect size of 0.26.

Table 1 shows the baseline characteristics by treatment arm. Mean baseline Tinnitus Handicap Inventory (THI) total scores were 48.5 (SD 19.5) in the single treatment group and 47.4 (SD 19.9) in the combined treatment group. Except for age and hearing aid indication, the baseline characteristics were generally well balanced between the treatment arms (see Table 1 and Table S6). Both age and hearing aid indication were considered as covariates during statistical analyses.

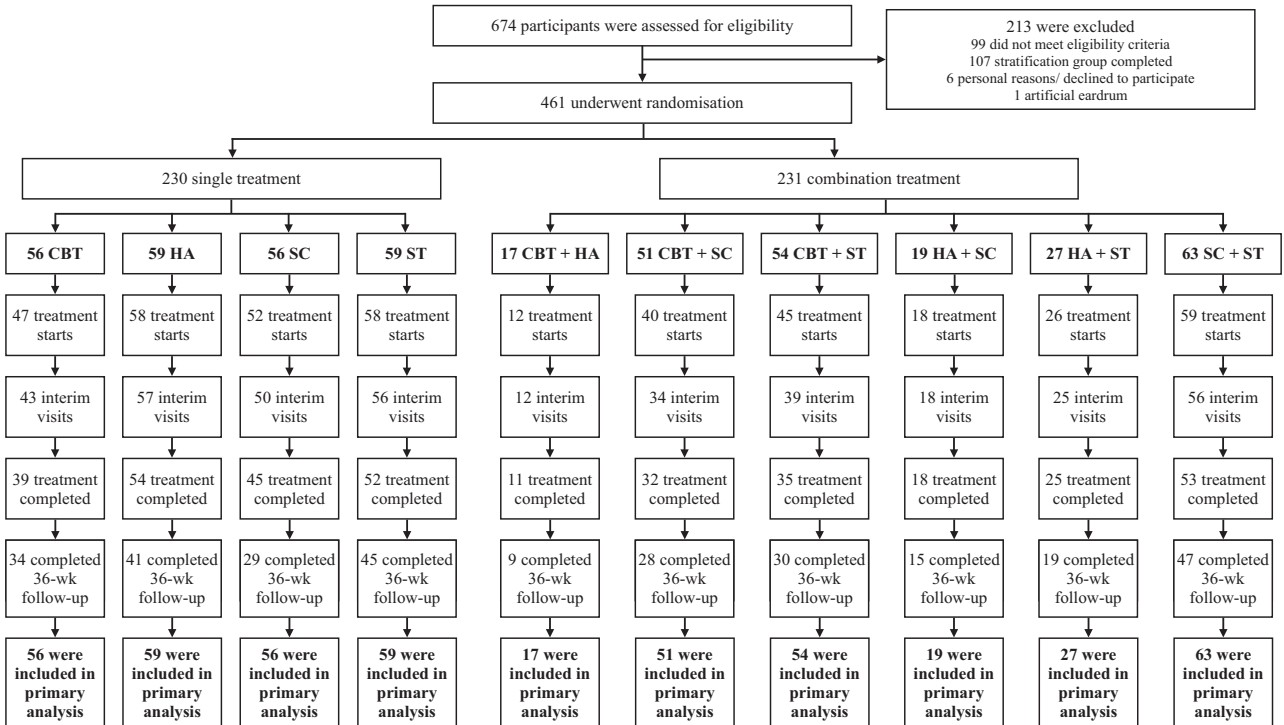

**Fig. 1 | Trial profile.** A total of 674 patients were screened, of whom 461 met the trial inclusion criteria and were randomly assigned to one of ten treatment arms comprised of a single treatment or a combination of two treatments out of four different therapy approaches - cognitive-behavioural therapy (CBT), hearing aids (HA), app-based structured counselling (SC) and app-based sound therapy (ST). 230 (49.9%) were assigned to single treatments (CBT, HA, SC, or ST) and 231 (50.1%) were assigned to combination treatments (CBT + HA, CBT + SC, CBT + ST, HA + SC, HA + ST, SC + ST). Patients without hearing aid indication were only randomised to treatments without HA. An extended version of the patient's flowchart can be found in Fig. S1. Quantity and reasons for trial exclusion during eligibility assessments and trial discontinuation/dropouts can be seen from Tables S1–S5.

**Table 1 | Demographic and clinical characteristics of the participants at baseline (stratified by treatment arm)**

| Characteristics | CBT (n = 56) | HA (n = 59) | SC (n = 56) | ST (n = 59) | CBT + HA (n = 17) | CBT + SC (n = 51) | CBT + ST (n = 54) | HA + SC (n = 19) | HA + ST (n = 27) | SC + ST (n = 63) | Overall (N = 461) |
|---|---|---|---|---|---|---|---|---|---|---|---|
| **Demographic characteristics** | | | | | | | | | | | |
| Sex | | | | | | | | | | | |
| Male (%) | 34 (60.7%) | 36 (61.0%) | 39 (69.6%) | 32 (54.2%) | 12 (70.6%) | 27 (52.9%) | 33 (61.1%) | 12 (63.2%) | 18 (66.7%) | 28 (44.4%) | 271 (58.8%) |
| Female (%) | 22 (39.3%) | 23 (39.0%) | 17 (30.4%) | 27 (45.8%) | 5 (29.4%) | 24 (47.1%) | 21 (38.9%) | 7 (36.8%) | 9 (33.3%) | 35 (55.6%) | 190 (41.2%) |
| Age (years) | 48.8 ±12.3 | 53.4 ±11.7 | 49.8 ±13.1 | 50.3 ±14.0 | 56.0 ±10.4 | 54.0 ±12.0 | 46.4 ±12.9 | 51.6 ±14.0 | 55.0 ±11.2 | 51.2 ±9.8 | 51.1 ±12.4 |
| PHQ-9 total score | 7.3 ±4.9 | 7.3 ±4.8 | 7.2 ±4.5 | 8.5 ±5.2 | 5.8 ±4.6 | 6.8 ±4.3 | 7.9 ±5.0 | 6.8 ±3.2 | 7.0 ±5.6 | 7.0 ±5.5 | 7.3 ±4.9 |
| **Tinnitus characteristics** | | | | | | | | | | | |
| Tinnitus duration (in months) | 119 ±127 | 126 ±100 | 85 ±77 | 115 ±114 | 101 ±111 | 154 ±140 | 110 ±99 | 159 ±144 | 124 ±108 | 119 ±116 | 119 ±113 |
| Hearing aid indication (%) | 19 (33.9%) | 59 (100%) | 19 (33.9%) | 20 (33.9%) | 17 (100%) | 18 (35.3%) | 17 (31.5%) | 19 (100%) | 27 (100%) | 19 (30.2%) | 234 (50.8%) |
| THI total score | 47.8 ±20.3 | 48.8 ±19.2 | 48.6 ±20.6 | 48.7 ±18.1 | 42.2 ±18.9 | 45.5 ±18.9 | 48.0 ±19.3 | 52.2 ±21.9 | 50.1 ±20.1 | 47.2 ±20.9 | 48.0 ±19.7 |
| TFI total score | 47.8 ±21.4 | 50.6 ±18.8 | 48.5 ±20.7 | 50.9 ±18.1 | 46.1 ±18.9 | 42.9 ±18.8 | 47.4 ±22.7 | 51.7 ±21.3 | 54.5 ±21.4 | 48.1 ±20.9 | 48.6 ±20.3 |
| Mini-TQ total score | 11.4 ±5.2 | 12.2 ±4.6 | 11.8 ±5.4 | 12.5 ±5.0 | 10.7 ±4.0 | 11.2 ±5.0 | 12.3 ±4.6 | 11.9 ±5.2 | 12.3 ±6.0 | 12.0 ±5.2 | 11.9 ±5.0 |
| NRS-tinnitus loudness | 6.2 ±2.1 | 6.7 ±1.7 | 6.4 ±2.4 | 6.3 ±2.1 | 6.3 ±2.7 | 6.0 ±2.6 | 6.2 ±2.6 | 6.4 ±2.3 | 7.2 ±1.6 | 6.3 ±2.2 | 6.4 ±2.2 |

Data are *n* (%) or mean ± SD. PHQ-9 scores range from 0 to 27, with higher scores indicating greater severity of depression. The definition for hearing aid indication is given in Table S37. THI scores range from 0 to 100, with higher scores indicating greater severity of tinnitus. TFI scores range from 0 to 100, with higher scores indicating greater severity of tinnitus. Mini-TQ scores range from 0 to 24, with higher scores indicating greater severity of tinnitus. NRS-tinnitus loudness scores range from 0 to 10, with higher scores indicating greater loudness of tinnitus.

*CBT* cognitive-behavioural therapy, *HA* hearing aids, *PHQ-9* Patient Health Questionnaire for Depression, *SC* app-based structured counselling, *ST* app-based sound therapy, *TFI* Tinnitus Functional Index, *THI* Tinnitus Handicap Inventory, *TQ* Tinnitus Questionnaire, *NRS* Numeric Rating Scale.

The difference in hearing aid indication results from randomising only individuals with relevant hearing loss to HA treatment arms. Results of audiometric measurements are shown in Figs. S2 and S3. Participants' baseline characteristics were similar to the group of persons with tinnitus seeking medical help in the general population (Table S7).

Regarding the primary objective, the least-squares mean change from baseline to week 12 in the THI total score was −11.7 (95% CI −14.4 to −9.0) for the single treatment groups and −14.9 (95% CI, −17.7 to −12.1) for the combination treatment arms (see Fig. 2 and Table 2) (interaction effect [single vs. combination treatments at final visit vs. baseline] $\beta = 3.2$, 95% CI, 0.2 to 6.1, $p = 0.034$).

Model parameters and model assumptions for the primary objective can be found in Table S8 and Fig. S4. The least-squares mean change from baseline to week 12 in the THI total score for the single vs. combination treatment comparison for each treatment strategy is reported in Table 2, and separately for every treatment arm in Table 3 and Fig. S5; and further separated by hearing aid indication in Table S9 and tinnitus severity in Table S10. Fig. 2 shows least-squares mean changes from baseline to interim visit at week 6, final visit at week 12 and follow-up at week 36 for both the overall and individual single-combination treatment comparison. The results of the remaining objectives (as outlined in the Statistical Analysis Plan (SAP))[17] and time points (interim visit and follow up) are reported in Tables S11–S13. Country-specific changes for the THI from baseline to final visit for single and combination treatment as well as for all treatment arms can be found in Table S14.

Regarding the secondary outcome measures, least-squares mean change from baseline to week 12 for Tinnitus Functional Index (TFI), Mini-TQ, PHQ-9, WHO-QoL and Numeric Rating Scales (NRS) (all objectives) are shown in Tables 2 and 3 as well as Tables S15–S27. Results of Clinical Global Impression Scale-Improvement (CGI-I) are reported descriptively for single and combination treatment groups at final visit, see Figs. S6 and S7, and separated by hearing aid indication (Fig. S8) and tinnitus severity (Fig. S9).

No Serious adverse event (SAE) was evident in any patient. Adverse events (AE) appeared in 49 (21.3%) participants in single treatment groups, and in 49 (21.2%) participants in combination treatment groups. The most relevant AEs reported by patients were worsening of the tinnitus percept (6); worsening of their psychological health (3); sleep problems (2); pain in the ear when wearing the hearing aid (1), ear infection (1), inflammation of the ear (1), dizziness (1) and mild transient hearing loss (1). Worsening of tinnitus symptoms is a relative common side-effect in tinnitus studies, as patients are focussing their attention more intently on their tinnitus to evaluate potential changes in tinnitus characteristics. Given the absence of any SAE and the low number of adverse reactions associated potentially with the various treatments, the present intervention types can be considered as safe. As AEs were rather rare and not severe, we abstained from analysing the strength of the relationship with treatment interventions and from documenting the time course of the reported AEs. A full listing of all AEs per treatment arm is provided in Table S28. Information on treatment adherence is given in Fig. S1 and Table S29.

Pairwise post-hoc contrasts for the THI least-squares mean change revealed statistically significant (Bonferroni adjusted) differences between ST and CBT, ST and CBT + SC, ST and CBT + ST, ST and HA, and ST and HA + SC. For all other treatment contrasts, no statistically significant differences were found (all $p > 0.050$). Statistical parameters for all post-hoc contrasts are listed in Table S30. Sensitivity analyses of our primary outcome using no imputation and the method of Last Observation Carried Forward yielded similar results as our intention-to-treat (ITT) analysis. However, under the assumption that data is not missing at random, our ITT findings cannot be upheld (Tables S31–S32). Per-protocol (PP) findings were different for the overall single vs. combination contrast (no statistical superiority of combination treatment; $\beta = 2.8$, 95% CI, −1.6 to 7.2, $p = 0.206$) (Fig. S10, Tables S33–S34). Exploratory analysis included the effect size estimates Cohen's d for all treatment arms which are shown in Table 3 and Fig. 2.

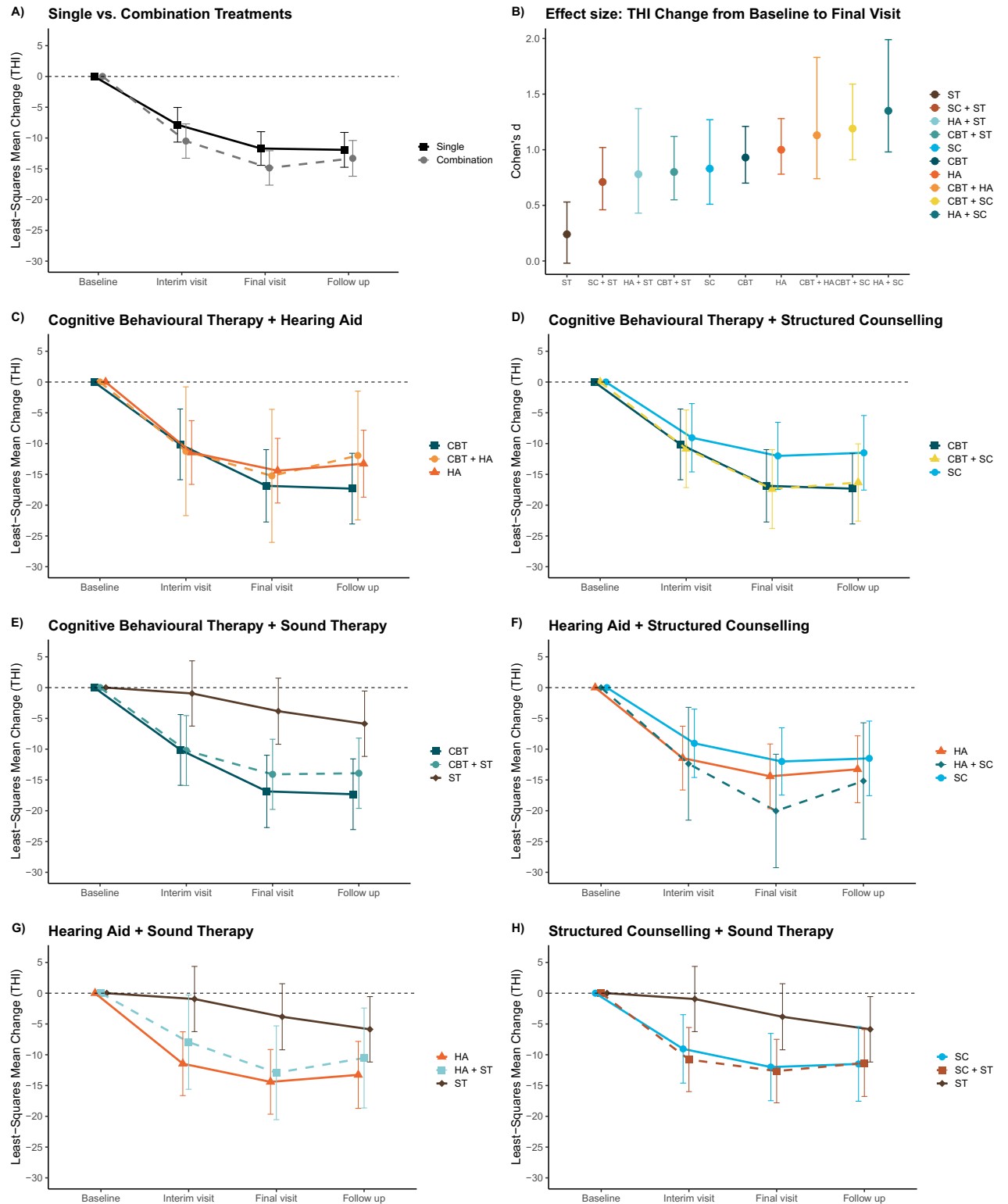

**Fig. 2 | Least-squares mean changes from baseline to interim visit (6w), final visit (12w) and follow-up (36w) in THI total score.** **A** single (*n* = 230) and combination (*n* = 231) treatments; **C** CBT + HA (*n* = 17); **D** CBT + SC (*n* = 51); **E** CBT + ST (*n* = 54); **F** HA + SC (*n* = 19); **G** HA + ST (*n* = 27); **H** SC + ST (*n* = 63); and **B** Cohen's *d* values for all treatment arms (change in THI total score from baseline to final visit).

Single treatment arms included: CBT (*n* = 56), HA (*n* = 59), SC (*n* = 56) and ST (*n* = 59). Total THI scores range from 0 to 100, with higher scores indicating greater severity of tinnitus. Error bars represent 95% confidence intervals. Abbreviations: CBT cognitive-behavioural therapy, HA hearing aids, SC app-based structured counselling, ST app-based sound therapy, THI Tinnitus Handicap Inventory.

**Table 2 | Primary and secondary clinical outcomes at final visit: single vs. combination (intention-to-treat population)**

| | All treatments | | Cognitive Behavioural Therapy | | Hearing Aid | | Structured Counselling | | Sound Therapy | |
|---|---|---|---|---|---|---|---|---|---|---|
| | Single | Combination | Single | Combination | Single | Combination | Single | Combination | Single | Combination |
| **Primary outcome** | | | | | | | | | | |
| THI | | | | | | | | | | |
| Change from baseline | −11.7 | −14.9 | −16.9 | −15.6 | −14.4 | −15.7 | −12.0 | −15.5 | −3.8 | −13.2 |
| (95% CI) | (−14.4 to −9.0) | (−17.7 to −12.1) | (−22.8 to −10.9) | (−19.5 to −11.7) | (−19.5 to −9.4) | (−20.7 to −10.7) | (−17.5 to −6.5) | (−19.3 to −11.7) | (−9.3 to 1.6) | (−16.7 to −9.8) |
| **Secondary Outcome** | | | | | | | | | | |
| TFI | | | | | | | | | | |
| Change from baseline | −11.0 | −11.6 | −16.1 | −12.1 | −14.5 | −13.9 | −9.7 | −10.1 | −3.7 | −11.7 |
| (95% CI) | (−13.9 to −8.0) | (−14.7 to −8.5) | (−22.1 to −10.1) | (−16.3 to −7.9) | (−20.2 to −8.9) | (−19.4 to −8.4) | (−15.5 to −3.9) | (−14.0 to −6.2) | (−9.6 to 2.1) | (−15.5 to −7.9) |
| Mini-TQ | | | | | | | | | | |
| Change from baseline | −2.9 | −3.4 | −4.1 | −3.8 | −3.5 | −3.0 | −2.9 | −3.4 | −1.2 | −3.0 |
| (95% CI) | (−3.6 to −2.2) | (−4.1 to −2.7) | (−5.5 to −2.6) | (−4.8 to −2.8) | (−4.7 to −2.4) | (−4.2 to −1.9) | (−4.3 to −1.4) | (−4.3 to −2.5) | (−2.6 to 0.2) | (−3.9 to −2.2) |
| NRS-tinnitus loudness | | | | | | | | | | |
| Change from baseline | −0.8 | −0.8 | −0.5 | −0.8 | −1.4 | −0.8 | −0.8 | −0.7 | −0.3 | −0.8 |
| (95% CI) | (−1.2 to −0.4) | (−1.2 to −0.4) | (−1.4 to 0.3) | (−1.4 to −0.2) | (−2.2 to −0.6) | (−1.6 to −0.1) | (−1.6 to 0.0) | (−1.2 to −0.2) | (−1.0 to 0.5) | (−1.3 to −0.3) |
| PHQ-9 | | | | | | | | | | |
| Change from baseline | −1.7 | −1.4 | −1.7 | −1.7 | −2.3 | −1.5 | −1.7 | −1.3 | −0.8 | −1.3 |
| (95% CI) | (−2.3 to −1.0) | (−2.1 to −0.8) | (−3.0 to −0.3) | (−2.6 to −0.8) | (−3.5 to −1.2) | (−2.6 to −0.4) | (−3.1 to −0.4) | (−2.2 to −0.5) | (−2.2 to 0.6) | (−2.2 to −0.4) |

Values depict least-squares mean changes at week 12 for primary and secondary outcomes with 95% confidence intervals. Higher total scores on the THI, TFI and Mini-TQ indicate greater severity of tinnitus. Higher total scores on the NRS - tinnitus loudness indicate greater loudness of tinnitus. Higher total scores on the PHQ-9 indicate greater severity of depression. Further objectives and secondary clinical outcomes not reported in this table can be seen in the Supplementary Appendix.

*NRS* numeric rating scale, *PHQ-9* Patient Health Questionnaire for Depression, *TFI* Tinnitus Functional Index, *THI* Tinnitus Handicap Inventory, *TQ* Tinnitus Questionnaire.

## Discussion

In this randomised trial on chronic tinnitus, the effectiveness of established tinnitus treatments (cognitive-behavioural therapy (CBT), hearing aids (HA), app-based structured counselling (ST) and app-based sound therapy (ST)) applied either alone or as a combination of two treatments was investigated. All treatments were safe and the improvement in THI scores from baseline to week 12 was statistically stronger for combination compared to single treatment. However, a more detailed analysis of our data by pairwise post hoc comparisons of the various treatment arms suggests that the additional effect of a treatment combination depends on the effectiveness of a single treatment. In the case of ST, a clear superiority in favour of combination treatment was present, with the combination CBT + ST being statistically superior to single ST. Importantly, there was no statistically significant difference between CBT alone and CBT + ST. This finding shows that combining a treatment with low effectiveness (in this case ST) together with a treatment of high effectiveness (in this case CBT) does not lead to a simple regression to the mean.

Rather the high-effectiveness treatment counterbalances the effect of the low-effectiveness treatment and elevates the clinical improvement up to a level comparable to the single high-effectiveness treatment. Together with the observation that ST was the treatment which demonstrated the smallest improvements in tinnitus-related handicap (statistically significant less than CBT, HA, CBT + SC, CBT + ST, HA + SC), the additional beneficial effect of a treatment combination appears to depend on how effective a single treatment already performs. For the single treatment arm with ST, we observed a weak effect size of 0.24 (confidence interval [CI], −0.02 to 0.53) while combinations of treatments including ST yielded medium to strong effect sizes: SC + ST (Cohen's $d = 0.71$, CI, 0.46 to 1.02), HA + ST (Cohen's $d = 0.78$, CI, 0.43–1.37) and CBT + ST (Cohen's $d = 0.80$. CI, 0.55–1.12), which is driven by the combination treatments of higher effectiveness.

The weak clinical effectiveness of sound treatment alone is in line with previous work where sound treatment was used as an active control[18,19]. This trial shows that combining a treatment of weak clinical effectiveness with a treatment of stronger clinical effectiveness counterbalanced the weak effect and provokes a clinical improvement comparable to the stronger effect. On the other hand, if a single treatment is already effective, a combination might not result in a synergistic effect.

Previous investigations evaluated combination treatments for tinnitus as well[9–11]. For instance it was demonstrated that Tinnitus Retraining Therapy[13], which combines a specific acoustic therapy with directive counselling, reduced tinnitus symptoms more effectively than counselling alone[9].

This is the first systematic trial to investigate CBT, HA, ST and SC within the scope of one investigation. CBT approaches demonstrate the best body of evidence so far and are thus recommended by current treatment guidelines[7,8,20]. Of today, the recommendation for HAs is restricted to the treatment of concomitant hearing loss and there is no recommendation for ST due to a lack of clear scientific evidence[21–23]. Counselling is recommended in form of information about tinnitus and the learning of potential coping strategies. However, counselling is usually not systematically structured and not investigated as such[24].

With the present trial, we can directly put into perspective the effect size of CBT as the most established evidence-based treatment in tinnitus[7,8,20,25,26], with HA, ST and SC (ST and SC provided with mobile applications) as well as their combinations as treatment options for tinnitus. Further, the present trial provides the first large-scale evidence for HA and SC (administered as stand-alone treatments), with a clinical effectiveness on a similar level as CBT.

In view of the interpretation of the present findings for HAs, it is important to point out that the primary focus of a HA is on reducing hearing impairment by amplification of peripheral sounds and that this

**Table 3 | Primary and secondary clinical outcomes at final visit: all treatment arms (intention-to-treat population)**

| | CBT | HA | SC | ST | CBT + HA | CBT + SC | CBT + ST | HA + SC | HA + ST | SC + ST |
|---|---|---|---|---|---|---|---|---|---|---|
| **Primary outcome** | | | | | | | | | | |
| THI | | | | | | | | | | |
| Change from baseline | −16.9 | −14.4 | −12.0 | −3.8 | −15.2 | −17.4 | −14.1 | −20.0 | −12.9 | −12.7 |
| (95% CI) | (−22.7 to −11.0) | (−19.7 to −9.2) | (−17.5 to −6.5) | (−9.2 to 1.5) | (−26.0 to −4.4) | (−23.8 to −11.0) | (−19.8 to −8.4) | (−29.3 to −10.8) | (−20.5 to −5.3) | (−17.8 to −7.5) |
| Cohen's d (95% CI) | 0.93 (0.70 to 1.21) | 1.00 (0.78 to 1.28) | 0.83 (0.51 to 1.27) | 0.24 (−0.02 to 0.53) | 1.13 (0.74 to 1.83) | 1.19 (0.91 to 1.59) | 0.80 (0.55 to 1.12) | 1.35 (0.98 to 1.99) | 0.78 (0.43 to 1.37) | 0.71 (0.46 to 1.02) |
| **Secondary Outcome** | | | | | | | | | | |
| TFI | | | | | | | | | | |
| Change from baseline | −16.1 | −14.5 | −9.7 | −3.7 | −15.1 | −10.9 | −12.2 | −10.1 | −15.8 | −9.4 |
| (95% CI) | (−22.2 to −10.0) | (−20.1 to −8.9) | (−15.6 to −3.8) | (−9.5 to 2.0) | (−26.1 to −4.0) | (−17.4 to −4.4) | (−18.5 to −5.9) | (−20.0 to −0.2) | (−24.0 to −7.6) | (−15.1 to −3.8) |
| Mini-TQ | | | | | | | | | | |
| Change from baseline | −4.1 | −3.5 | −2.9 | −1.2 | −4.0 | −4.1 | −3.6 | −3.2 | −2.3 | −2.9 |
| (95% CI) | (−5.5 to −2.6) | (−4.8 to −2.2) | (−4.3 to −1.4) | (−2.6 to 0.2) | (−6.7 to −1.3) | (−5.6 to −2.6) | (−5.0 to −2.2) | (−5.5 to −0.9) | (−4.3 to −0.4) | (−4.2 to −1.6) |
| NRS - tinnitus loudness | | | | | | | | | | |
| Change from baseline | −0.5 | −1.4 | −0.8 | −0.3 | −1.0 | −0.9 | −0.7 | −0.3 | −1.1 | −0.7 |
| (95% CI) | (−1.4 to 0.3) | (−2.1 to −0.6) | (−1.6 to 0.0) | (−1.1 to 0.5) | (−2.5 to 0.6) | (−1.8 to 0.0) | (−1.5 to 0.1) | (−1.6 to 1.1) | (−2.2 to -0.1) | (−1.5 to 0.0) |
| PHQ-9 | | | | | | | | | | |
| Change from baseline | −1.7 | −2.3 | −1.7 | −0.9 | −1.2 | −1.8 | −1.8 | −2.0 | −1.3 | −0.8 |
| (95% CI) | (−3.0 to −0.3) | (−3.6 to −1.1) | (−3.1 to −0.4) | (−2.2 to 0.4) | (−3.7 to 1.2) | (−3.2 to −0.3) | (−3.2 to −0.4) | (−4.2 to 0.2) | (−3.1 to 0.6) | (−2.1 to 0.4) |

Values depict least-squares mean changes at week 12 for primary and secondary outcomes with 95% confidence intervals. Higher total scores on the THI, TFI and Mini-TQ indicate greater severity of tinnitus. Higher total scores on the NRS - tinnitus loudness indicate greater loudness of tinnitus. Higher total scores on the PHQ-9 indicate greater severity of depression. Cohens d indicate the standardised effect size of the respective treatment. The effect sizes and the corresponding confidence intervals were first computed in each of the 50 imputed data sets before they were averaged to a single value. Further objectives and secondary clinical outcomes not reported in this table can be seen in the Supplementary Appendix.

*CBT* cognitive-behavioural therapy, *HA* hearing aids, *NRS* numeric rating scale, *PHQ-9* Patient Health Questionnaire for Depression, *SC* app-based structured counselling, *ST* app-based sound therapy, *TFI* Tinnitus Functional Index, *THI* Tinnitus Handicap Inventory, *TQ* Tinnitus Questionnaire.

benefit could be conflated with an amelioration in tinnitus-related symptoms.

In a separate analysis by Schiele et al., data from our HA single treatment arm was used to investigate whether tinnitus frequency, hearing loss, HA-usage duration or the accuracy of HA fitting might serve as a predictor for treatment response. None of the mentioned variables predicted an improvement in tinnitus-related distress (THI, TFI) or subjective tinnitus loudness[27].

The combination of HA + SC, which provided the strongest effect size in our trial, has not been investigated so far, and data about the clinical effectiveness in tinnitus are not yet available[21,22]. It should also be considered that we worked with a selected set of four tinnitus treatments and combinations of only two treatment types. Thus, it remains unknown, whether the combination of other treatment sets or combinations of three or more treatment types would lead to additional treatment benefits. Any interpretation of our findings should keep in mind, that we investigated specific applications of CBT, HA, ST and SC. Potential reasons for the low efficacy of ST and SC in the present trial might include its self-administration, the limited interaction with a clinical specialist and/or the absence of specific instructions (stimulus, loudness, duration etc.). Thus, our conclusions on ST and SC might not be directly applied to a traditional clinical setting, where patients are not necessarily followed-up.

The duration of treatment was 12 weeks in all treatment arms. Meaningful clinical improvements were observed in most treatment arms after 6 weeks and improved further towards the final assessment after 12 weeks and remained during the follow-up period.

Despite the usage of interventions allowing for a high level of patient flexibility (SC and ST via mobile applications, HA), treatment compliance/adherence was low (see Fig. S1 and Table S29) and dropout rates were high in our trial (per-protocol (PP) sample of 185 patients).

CBT treatment arms, which require a high level of commitment with several on-site visits, demonstrated the highest proportion of dropouts in our trial, which potentially limits the interpretability and robustness of our CBT findings, as non-responders may be overrepresented among dropouts. In another recent study, in which CBT was compared with Neurofeedback, the CBT dropout rate was in a similar high range like in our study[28]. There is a large body of evidence in the literature that CBT is effective in the treatment of tinnitus (for an overview see the Cochrane review by Fuller et al.)[25], and has been recommended in European guidelines for the management of tinnitus[8]. However, all studies investigating CBT alone might be susceptible to a selection bias, as only patients with motivation for CBT would have been enroled. The relatively high dropout rate of CBT in studies comparing various treatment options reflects the clinical experience of the real-world situation where a relevant subgroup of patients is not willing to undergo CBT. Detailed information on dropout reasons per treatment arm are listed in Tables S2 – S5.

With the application of two treatments in combination, the chances that one or even both treatments are not conducted as intended are increasing. The lack of monitoring, strict guidance, or outpatient care in the case of SC, ST and HA, might be further potential reasons for treatment non-adherence. Furthermore, high dropout rates are a well-known issue in mobile health interventions[29]. Another reason could be that patients were randomised to treatments and did not receive the treatment they desired. Under ideal treatment

compliance/adherence (PP analysis), we observed no overall superiority of combination treatments.

A potential explanation for this incongruency between ITT and PP analysis might be that under perfect conditions (PP), a single treatment which is conducted properly is already effective on its own and thus there is no clear additional beneficial effect of a combination treatment. However, if one or two treatments are not properly conducted (ITT), as it is most probably the case in the everyday clinical treatment of tinnitus, a combination of treatments provides an additional benefit. Our results indicate that there is a high need for further research to better understand the clinical benefits of combination treatment; to get more profound insights behind the reasons for low treatment adherence; and in approaches to increase treatment adherence in daily clinical practice, such as the implementation of behavioural change techniques or more extensive patient education.

A control group was not included in this trial, as the answer to the main question (comparison of single and combined treatment) did not require a control group. Nevertheless, a control group may have been helpful as an anchor for comparison with the ten treatment arms. However, our results of CBT as single treatment correspond very well to meta-analytic data of its efficacy[25] and thus provide an anchor for a well-established evidence based treatment approach. Further, our data demonstrates low effectiveness of ST as a single treatment, supporting its use as an active control condition in randomised controlled trials[18,19]. Thus, the two treatment arms CBT and ST can be considered as reliable reference anchors for the interpretation of the results of the other 8 investigated treatment arms. Even though in 18% of all participants data of the primary outcome (THI) was missing, the sensitivity analysis using no imputation came to similar findings, which was further corroborated by applying the Last Observation Carried Forward approach. Yet, under the assumption of 'missing not at random' and after conducting additional robustness evaluations using three different reference-based imputation methods, our findings cannot be sustained (see Tables S31–S32).

In this trial involving adults with chronic tinnitus, we found that 12 weeks of treatment with CBT, HAs, SC, or ST applied as single or in combinations of two treatments led to an amelioration in tinnitus-related handicap. There was no unambiguous synergistic effect of treatment combination, rather a compensatory effect, where a more effective treatment offsets the clinical effects of a less effective treatment. In clinical situations where it is unclear which treatment will benefit the particular patient, a combination of treatments might help to increase the chances of treatment success.

## Methods

### Study design

This was an investigator-initiated, international, multicentre, parallel-arm, superiority, randomised controlled clinical trial conducted in five hospitals across four European countries (Leuven, Belgium; Berlin and Regensburg, Germany; Athens, Greece; and Granada, Spain; see Table S35 in the Supplementary Appendix) as part of the UNITI project (Unification of Treatments and Interventions for Tinnitus Patients)[30]. Included patients received treatment between April 2021 and December 2022. Detailed information about the trial rationale, design, methodological approaches and statistical analysis strategies are published in the study protocol and statistical analysis plan (SAP)[16,17]. The study was approved by local ethics committees at every clinical site independently (combined ethical approval for German sites; please find the ethical approval documents in the Supplementary Appendix). Further, all authors vouch for the completeness and correctness of the data, adherence of the trial to the study protocol[16], as well as adherence of data analysis strategies to the SAP[17]. A detailed list of author contributions can be found in Table S36 in the Supplementary Appendix. Written informed consent was obtained from all eligible patients prior to trial participation. For the preparation of this report we used the CONSORT guidelines (Consolidated Standards of Reporting Trials)[31].

### Participants

Adults of both sexes (self-reported) aged between 18 and 80 years with chronic subjective tinnitus (lasting for 6 months or more) were recruited and screened at each clinical site. Inclusion criteria for trial participation were at least mild tinnitus handicap according to the Tinnitus Handicap Inventory[32] (THI; score ≥ 18) and tinnitus as primary complaint. Exclusion criteria were: presence of a mild or worse cognitive impairment according to the Montreal Cognitive Assessment[33] (MoCa; score ≤ 22); any relevant ear disorders or acute infections of the ear; one deaf ear; severe hearing loss (inability to communicate properly) as well as serious internal, neurological, or psychiatric conditions. Existing drug therapies with psychoactive substances had to be stable, and no start of any other tinnitus-related treatment in the last 3 months before trial participation was allowed. A detailed list of all eligibility criteria can be found in the trial protocol[16]. Written informed consent was obtained from all participants.

### Randomisation and blinding

After successful on-site screening, eligible participants were stratified in four equally sized strata based on their THI total score (low [<48] and high [≥48] tinnitus-related handicap) and hearing aid indication (yes and no, criteria for hearing aid indication: Table S37). Criterion for low and high tinnitus-related handicap was defined based on historical data obtained from 837 patients at the clinical site in Regensburg with a median THI score of 48. Hearing aid indication criteria were specified by a group of international experts in the fields of audiology and otolaryngology (see Table S38). Participants were then randomised to one of ten treatment arms comprised of single (CBT, HA, SC, ST) and combination interventions (CBT + HA, CBT + SC, CBT + ST, HA + SC, HA + ST, SC + ST) under consideration of the stratification group. Patients from the two strata without hearing aid indication were not randomised in treatment groups that comprised HA treatment. The stratification according to tinnitus-related handicap was performed to ensure an equal representation of patients with high and low tinnitus distress in different treatment arms and thus avoid potential misinterpretations of our findings due to large differences in baseline tinnitus severity across treatment arms. Randomisation was conducted at each clinical site with an interactive web response system developed together with biostatisticians from the contract research organisation Excelya (www.excelya.com). Excelya was further responsible to monitor all randomisation proceedings. Treatment codes were used to ensure blindness of the statistical analysis team to the type of treatment patients received. Unblinding was conducted after analyses completion. Patients and investigators/assessors were not blinded. See study protocol and statistical analysis plan for more detailed information[16,17].

### Procedures

Single and combination treatments were applied over a 12-week treatment phase. All treatment procedures were designed by dedicated experts in their respective fields (see Table S38 for expert team per treatment type) and described in detail in the study protocol[16]. To ensure consistency with respect to treatment and assessment implementation across clinical sites, workshops were held and Standard Operation Procedure documents were created. Two of the four treatment types were unguided, app-based therapies, minimising potential differences. HA fitting was standardised, and CBT was co-developed specifically for this trial.

CBT was based on the concept of fear-avoidance using exposure therapy[34,35]. The exposure exercises were delivered by trained psychologists or psychotherapists in weekly face-to-face group sessions

(1.5−2 h weekly; 12 weeks; group size: six to eight participants). For HA treatment, behind-the-ear hearing instruments (Type Signia Pure 312 7X; Sivantos Pte. Ltd., Singapore, Republic of Singapore/WSAudiology, Lynge, Denmark) were fitted bilaterally with all noise-related signal processing deactivated by audiologists or HA acousticians according to the National Acoustic Laboratories-Non-Linear 2 generic amplification proceeding[36]. SC and ST were self-administered on a daily basis via a dedicated UNITI mobile application, which was available for Android and iOS devices as well as free of charge[37]. SC was oriented on recent European guidelines for tinnitus management[8] and consisted of 12 chapters featuring structured patient education (e.g. facts about tinnitus, brain and sound perception; myths and misconceptions about tinnitus; diagnosis of tinnitus; special types of tinnitus; therapeutic approaches; psychological and behavioural aspects) and tips on how to handle tinnitus distress. ST included 64 different artificial and naturalistic sounds with various state of the art modulation or filter techniques. Loudness and length of the sounds was adjustable by the patients. There are many different SC and ST approaches administered by clinicians. For clarity, we want to mention that our app-based approach did not follow the Tinnitus Retraining Therapy protocol.

Treatment compliance was assessed via participation in CBT treatment sessions (≥6 CBT sessions; including the first two), usage log files for HAs (average use of ≥4 hours/day) and app-use logfiles for SC (completion of the first six chapters) and ST (using each of the four sound stimuli categories once)[17]. Demographic and clinical characteristics were assessed at baseline (before treatment) using the European School of Interdisciplinary Tinnitus Research Screening Questionnaire (ESIT-SQ)[38]. Outcome measures were assessed at baseline, interim (after 6 weeks of treatment), final (after 12-week treatment period) and follow-up (36 weeks after baseline) visits. An additional follow-up visit was conducted 48 weeks after baseline. This visit was a voluntary follow-up visit. Due to a large amount of missing data (only 32.54% of participating patients provided data), no reliable conclusions can be drawn from the analysis and therefore this additional follow-up was not included in the final outcome measure analysis.

## Outcome measures

The primary outcome between single and combination treatment was the difference in total score change from baseline to final visit (after 12 weeks of treatment) in the Tinnitus Handicap Inventory (THI)[32]. The THI consists of 25 items designed to evaluate the perceived impact of tinnitus on an individual's daily life. Each item provides three response options: 'No', 'Sometimes' and 'Yes', which are scored as 0.2 and 4 points respectively. The total THI score is obtained by summing the scores of all items, resulting in a score that ranges from 0 to 100, with higher sores indicating greater perceived handicap due to tinnitus. Changes from baseline to interim visit, and follow-up were examined in secondary analyses as well. Despite some critique on its sensitivity[39,40], the THI was chosen as the primary outcome measure, since (i) it is the most widely used instrument in clinical settings and is recommended as an outcome for clinical trials based on expert consensus[41–43], (ii) there is high evidence of a conformity between the THI, the Tinnitus Functional Index (TFI)[44] and the Tinnitus Questionnaire (TQ)[45], plus (iii) a validated version was available in the required languages (Dutch, German, Greek, Spanish) at the time of trial registration and the definition of our primary outcome measure[46–49].

Secondary outcome measures included the TFI, the Mini Tinnitus Questionnaire (Mini-TQ)[50], the Patient Health Questionnaire for Depression (PHQ-D/PHQ-9)[51], the abbreviated version of the World Health Organisation−Quality of Life questionnaire (WHO-QoL)[52] as well as numeric rating scales (NRS; 0−10) for tinnitus impairment (0−not a problem; 10−very big problem), tinnitus loudness (0−not at all loud; 10−extremely loud), tinnitus-related discomfort (0−no discomfort; 10−severe discomfort), annoyance (0−not at all annoying; 10−extremely annoying), unpleasantness (0−not at all unpleasant;

10−extremely unpleasant), and ability to ignore the tinnitus (0−very easy to ignore; 10−impossible to ignore)[53]. Clinical improvement was measured with the Clinical Global Impression Scale-Improvement (CGI-I)[54]. There is expert-based consensus on which outcome domains should be ideally assessed in tinnitus trials. However, there is still no consensus-based recommendation on which standardised instruments should be used within the selected outcome domains[55]. Different secondary outcome measures were considered here to underpin interpretability, validity as well as comparability of potential findings with past and future research.

Questionnaires were filled out by the patients using a graphical interface of the UNITI database[16]. Patients could also opt for paper-pencil versions, and data was subsequently entered into the UNITI database by the local study team.

Adverse (AE) and serious adverse events (SAE) were defined according to the guidelines for Good Clinical Practice §3 (6,8). AEs were assessed and recorded during each visit with respect to start and end date, intensity, relation to intervention, impact on treatment and actions taken. Any SAE during the 12-week treatment phase led to a stop of the patient's respective treatment and was immediately reported to the local ethics committee.

## Statistical analysis

The sample size was determined a priori on an estimated effect size of 0.26, an alpha level of 5% and a power of 80% (two-sided test). Based on that, the necessary sample size is 468. Considering potential dropouts, the aim was to recruit a total sample size of $N = 500$[16].

The statistical analysis was performed in the ITT population of $N = 461$, including all randomised participants, regardless of compliance with the study protocol. For the primary analysis (combination against single treatments), we estimated that with a two-tailed alpha level of less than 0.05, the sample size of $N = 461$ provides the trial with 90% power to detect an effect size of 0.30 (lower end of 95% CI for effect size of behavioural therapy interventions according to the latest Cochrane Review on tinnitus)[25].

For the ITT analysis, missing values (THI: 18%, education: 3.5%, PHQ-9 baseline: 2.6%) were imputed using multilevel imputation (R package `mitml`)[56,57]; see Fig. S11 for the distribution of imputed THI values. This approach is considered the gold standard for dealing with missing data[58]. As sensitivity analysis, a per-protocol (PP) was conducted on all patients who met the requirements for treatment compliance as defined in the SAP ($N = 185$)[17]. Additional sensitivity analyses were performed in the primary outcome without imputation, three different reference-based imputation approaches (*jump to reference*, *copy increments in reference*, *copy reference*, R package `RefBasedMI`)[59,60] assuming data is not missing at random and the method of Last Observation Carried Forward. The analysis of the primary objective was performed in the ITT population to test the effectiveness of combination treatments against single treatments (control group). Further comparisons between single versus combination treatments for all 4 single treatments separately (CBT single vs. combined, HA single vs. combined, SC single vs. combined, ST single vs. combined) as well as comparisons between all ten treatment arms were performed. Detailed information on which treatment arms were pooled for which type of comparison can be found in the SAP[17].

To address all objectives, mixed effect models were applied (with REML using the `lme4` R package)[61] by considering the outcome as the response variable and including the corresponding objective, time point (baseline, interim visit, final visit and follow-up), and objective-by-time interaction as fixed effects, including centre and subject ID as random intercepts. The models were adjusted for the following covariates: age, sex, educational attainment, hearing aid indication and PHQ-9 baseline scores[17]. The results of the remaining objectives as described in the SAP are reported in the Supplementary Appendix. Additionally, we evaluated THI score changes from baseline to final

visit for single and combination treatment as well as all individual treatment arms separately by country to assess potential country-specific effects. Results are reported as least-squares mean changes (obtained via the emmeans R package)[62] with 95% CI. All analyses were performed in R (version 4.2.2).

De-identified data (pseudo-anonymised code) were gathered in a central database, which was regularly monitored and systematically checked for missing and invalid data (every 6 weeks). After database closure and prior to analysis, data from each clinical centre were checked again for validity and completeness. This study was registered at ClinicalTrials.gov, NCT04663828.

### Reporting summary
Further information on research design is available in the Nature Portfolio Reporting Summary linked to this article.

## Data availability
De-identified data analysed and reported in the article and Supplementary Appendix are available upon reasonable request from the corresponding author (stefan.schoisswohl@ukr.de). Source data are provided with this paper. The complete dataset (incl. 48-week follow-up) and its description is currently under preparation for publication and release via ZENODO. The current status of data availability will be updated on the UNITI website (https://uniti.tinnitusresearch.net/).

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

## Acknowledgements

We would like to thank all patients who participated in this trial, without whom this research would not have been possible. We would like to further thank the whole consortium of the UNITI-project for their feedback and support. Moreover, we would like to thank Simon Grund for his support regarding the mitml R package.

This clinical trial received funding from the European Union's Horizon 2020 Research and Innovation Program (grant agreement number: 848261). SSch received funding outside the present study from dtec.bw —Digitalization and Technology Research Centre of the Bundeswehr (MEXT project). dtec.bw is funded by the European Union—NextGenerationEU. M.E. received research funding outside the present study from the Rainwater Charitable Foundation and the Sonova Holding AG. A.E.B. received research funding outside the present study from ibs.Granada/ Fundación para la Investigación Biosanitaria de Andalucía Oriental (FIBAO), European Molecular Biology Organisation (EMBO) and Centro de Investigación Biomédica en Red Enfermedades Raras (CIBERER). A.G.M. received research funding outside the present study from the CECEU 2020, Andalusian Government of Spain (grant number: DOC_01677). P.P.C. received research funding outside the present study from the Consejería de Salud y Familias, Junta de Andalucía. 2020, Contrato Posdoctorales Especialistas (RH–0150–2020), and Instituto de Salud Carlos III., Bases neurofisiológicas y perfil de seguridad de la terapia sonora en pacientes con acúfeno crónico severo (PI22/01838). N.V. received research funding outside the present study from the Flanders Research Foundation and VLAIO—KU Leuven. The funders had no influence on trial design and had no role in the collection, analysis, interpretation of the data, preparation of the manuscript or in the decision to submit the manuscript for publication.

## Author contributions

Stefan Schoisswohl (S.Sch.) contributed to conceptualisation, investigation, formal analysis, methodology, project administration, supervision, visualisation and writing—original draft; Winfried Schlee (W.S.) contributed to conceptualisation, funding acquisition, formal analysis, methodology, project administration, supervision, visualisation and writing—original draft; Berthold Langguth (B.L.) contributed to conceptualisation, funding acquisition, methodology, supervision and writing—review and editing; Laura Basso (L.B.) and Milena Engelke (M.E.) contributed to data curation, formal analysis, methodology, visualisation and writing—original draft; Rüdiger Pryss (R.P.) contributed to funding acquisition, investigation, methodology, software and writing—review and editing; Birgit Mazurek (B.M.), Jose Antonio Lopez-Escamez (J.A.L.E.), Dimitrios Kikidis (D.K.) and Rilana Cima (R.C.) contributed to conceptualisation, funding acquisition, methodology and writing—review and editing; Myra Spiliopoulou (M.Sp.) contributed to funding acquisition, data curation, formal analysis and writing—review and editing;

Susanne Staudinger (S.Stau.) contributed to conceptualisation, project administration, investigation and writing—review and editing; Carsten Vogel (C.V.) contributed to investigation, methodology, software and writing—review and editing; Jorge Simoes (J.S.), Uli Niemann (U.N.), Clara Puga (C.P.), Miro Schleicher (M.Schl.), Carlotta M. Jarach (C.M.J.), Hafez Kader (H.K.) and Vishnu Unnikrishnan (V.U.) contributed to data curation, formal analysis, methodology and writing—review and editing; Benjamin Boecking (B.B.) and Martin Schecklmann (M.Sche.) contributed to conceptualisation, methodology, investigation and writing—review and editing; Christopher R. Cederroth (C.R.C.) contributed to funding acquisition, resources and writing—review and editing; Steven C. Marcrum (S.C.M.) contributed to conceptualisation, investigation and writing—review and editing; Patrick Neff (P.N.) contributed to conceptualisation, methodology and writing—review and editing; Johannes Schobel (J.S.) contributed to investigation, software and writing—review and editing; Alberto Bernal-Robledano (A.B.R.), Marta Martinez-Martinez (M.M.M.), Nicolas Muller-Locatelli (N.M.L.), Patricia Perez-Carpena (P.P.C.), Paula Robles-Bolivar (P.R.B.), Matthias Rose (M.R.), Tabea Schiele (T.S.), Sam Denys (S.D.), Alba Escalera-Balsera (A.E.B.), Alvaro Gallego-Martinez (A.G.M.), Leyre Hidalgo-Lopez (L.H.L.), Nikos Markatos (N.M.), Juan Martin-Lagos (J.M.L.), Sabine Stark (S.Sta.), Alexandra Stege (A.S.), Evgenia Vassou (E.V.), Nicolas Verhaert (N.V.) and Zoi Zachou (Z.Z.) contributed to investigation and writing—review and editing; Jan Bulla (J.B.) and Beat Toedtli (B.T.) contributed to formal analysis, validation and writing—review and editing; Silvano Gallus (S.G.) contributed to funding acquisition and writing—review and editing; Michael Koller (M.K.), Hazel Goedhart (H.G.) and Holger Crump (H.C.) contributed to conceptualisation and writing—review and editing; and Alessandra Lugo (A.L.) and Ilias Trochidis (I.T.) contributed to writing—review and editing. All authors had full access to all the data in the study and had final responsibility for the decision to submit for publication. S.Sch., W.S., L.B. and M.E. have directly accessed and verified the underlying data reported in the manuscript.

## Funding

## Competing interests

The authors declare no competing interests.

## Additional information

Stefan Schoisswohl [1,2] ✉, Laura Basso [1], Jorge Simoes[3], Milena Engelke[1], Berthold Langguth [1], Birgit Mazurek[4], Jose Antonio Lopez-Escamez [5,6,7], Dimitrios Kikidis [8], Rilana Cima[9], Alberto Bernal-Robledano[6,7], Benjamin Böcking [4], Jan Bulla[1,10], Christopher R. Cederroth [11,12], Holger Crump[13], Sam Denys [14,15], Alba Escalera-Balsera [6,7], Alvaro Gallego-Martinez [6,7], Silvano Gallus[16], Hazel Goedhart[17], Leyre Hidalgo-Lopez[18], Carlotta M. Jarach[16], Hafez Kader[19], Michael Koller[20], Alessandra Lugo[16], Steven C. Marcrum [21], Nikos Markatos[8], Juan Martin-Lagos[6,22], Marta Martinez-Martinez[6,22], Nicolas Müller-Locatelli[22], Patrick Neff[1,23], Uli Niemann[19], Patricia Perez-Carpena[6,7,22], Rüdiger Pryss [24,25], Clara Puga[19], Paula Robles-Bolivar[6,7], Matthias Rose[26], Martin Schecklmann[1], Tabea Schiele[4], Miro Schleicher[19], Johannes Schobel [27], Myra Spiliopoulou[19], Sabine Stark[4], Susanne Staudinger[1], Alexandra Stege[28], Beat Tödtli[29], Ilias Trochidis[30], Vishnu Unnikrishnan[19], Evgenia Vassou[8], Nicolas Verhaert[14,15], Carsten Vogel [24,25], Zoi Zachou[8] & Winfried Schlee[1,29]

[1]Department of Psychiatry and Psychotherapy, University of Regensburg, Regensburg, Germany. [2]Department of Human Sciences, Institute of Psychology, Universität der Bundeswehr München, Neubiberg, Germany. [3]Department of Psychology, Health and Technology, University of Twente, Enschede, The Netherlands. [4]Tinnitus Center, Charité—Universitätsmedizin Berlin, corporate member of Freie Universität Berlin, Humboldt-Universität zu Berlin, and Berlin Institute of Health, Berlin, Germany. [5]Meniere's Disease Neuroscience Research Program, Faculty of Medicine & Health, School of Medical Sciences, The Kolling Institute, University of Sydney, Sydney, NSW, Australia. [6]Otology & Neurotology Group CTS495, Instituto de Investigación Biosanitaria, ibs.GRANADA, Universidad de Granada, Granada, Spain. [7]Sensorineural Pathology Programme, Centro de Investigación Biomédica en Red en Enfermedades Raras, CIBERER, Madrid, Spain. [8]First Department of Otorhinolaryngology, Head and Neck Surgery, National and Kapodistrian University of Athens, Hippocrateion General Hospital, Athens, Greece. [9]Department of Health Psychology, Katholieke Universiteit Leuven, Leuven, Belgium. [10]Department of Mathematics, University of Bergen, Bergen, Norway. [11]Department of Physiology and Pharmacology, Karolinska Institutet, Stockholm, Sweden. [12]Translational Hearing Research, Tübingen Hearing Research Center, Department of Otolaryngology, Head and Neck Surgery, University of Tübingen, Tübingen, Germany. [13]Patient Organisation "Hast Du Töne—Leben mit Tinnitus" Bergisch-Gladbach, Bergisch-Gladbach, Germany. [14]Department of Otorhinolaryngology, Head & Neck Surgery,

University Hospital Leuven, Katholieke Universiteit Leuven, Leuven, Belgium. [15]Department of Neurosciences, Research Group Experimental Otorhinolaryngology (ExpORL), Katholieke Universiteit Leuven, Leuven, Belgium. [16]Department of Medical Epidemiology, Istituto di Ricerche Farmacologiche Mario Negri IRCCS, Milan, Italy. [17]Tinnitus Hub, London, UK. [18]Department of Mental Health, Hospital Universitario Virgen de las Nieves, Granada, Spain. [19]Faculty of Computer Science, Otto-von-Guericke Universität, Magdeburg, Germany. [20]Center for Clinical Studies, University of Regensburg, Regensburg, Germany. [21]Department of Otolaryngology, University Hospital Regensburg, Regensburg, Germany. [22]Department of Otolaryngology, Hospital Clinico Universitario San Cecilio, Granada, Spain. [23]Department of Otorhinolaryngology, Head&Neck Surgery, University Hospital Zurich, University of Zurich, Zurich, Switzerland. [24]Institute of Clinical Epidemiology and Biometry University of Würzburg, Würzburg, Germany. [25]Institute of Medical Data Science, University Hospital of Würzburg, Würzburg, Germany. [26]Department of Internal Medicine and Psychosomatics, Charité—Universitätsmedizin Berlin, corporate member of Freie Universität Berlin, Humboldt-Universität zu Berlin, and Berlin Institute of Health, Berlin, Germany. [27]Institute DigiHealth, Neu-Ulm University of Applied Sciences, Neu-Ulm, Germany. [28]Centrale Biobank Charité (ZeBanC), Charité—Universitätsmedizin Berlin, corporate member of Freie Universität Berlin, Humboldt-Universität zu Berlin, and Berlin Institute of Health, Berlin, Germany. [29]Institute for Information and Process Management, Eastern Switzerland University of Applied Sciences, St. Gallen, Switzerland. [30]ViLabs, Limassol, Cyprus. ✉e-mail: stefan.schoisswohl@ukr.de

