## [Transparent Peer Review file · Nature Communications]

Single versus Combination Treatment in Tinnitus: An International, Multicentre, Parallel-arm, Superiority, Randomised Controlled Trial

Corresponding Author: Dr Stefan Schoisswohl

Version 0:

Reviewer comments:

Reviewer #1

(Remarks to the Author)

The authors have performed a much needed and challenging clinical study in the tinnitus field that assessed several routine treatments either alone or in combination. It is not uncommon for many tinnitus patients to try multiple treatments for tinnitus; yet the efficacy of one or a series of treatments is not well understood. Although quite challenging to assess so many different treatments and combinations, the authors should be commended for pursuing an initial study to try to better understand what is pursued in real world clinics and performing a quite large study to have enough power to test a few comparisons. Overall, the paper is well written and clear, and rigorously done with pre-specified protocol/SAPS that were published beforehand. They were able to show that combo treatments can be effective but not necessarily better than single treatments; instead, more to offer alternative options in which the better performing treatment for a given patient is mainly what dictates their overall outcomes. That finding is still clinically impactful because it further supports that patients should still be provided different options and combinations since they have a chance of benefiting sufficiently based on one that is most effective for them. Furthermore, ST has proven to be minimally effective but can still be useful combined with other approaches, while HAs had the largest effect size (see comment below but not fully convincing if CBT should also be considered as also having large effect size).

There are a few suggestions provided below to potentially further help the clarity of the paper and its interpretation.

Four general concerns/considerations:

1) The study deliberately stratified patients into four equally sized strata based on THI total score (cutoff of 48) and hearing aid indication (yes/no). It was not clearly justified how the authors came up with these strata, so it would be helpful to better clarify why these strata were selected. More importantly, if so important to stratify based on those groups, then those results should be included in the main paper and some key interpretations/conclusions should be presented in the paper about them (only saw 2 tables lightly mentioned in the Supplementary document). Please provide additional plots or results in the main paper to interpret those findings or justify why they are no longer critical for the original design/expectations of the study.

2) CBT has been referred to in the paper (e.g., Discussion) as the most established treatment and in this current study there is an interpretation that it alone performed well (at least compared to ST). However, from Figure 1, it also had the largest drop-out rate. It isn't clear how multiple imputation or the current analyses can support the efficacy of CBT when so many have dropped out specifically for CBT (e.g., are those who dropped out non-responders or would ultimately cause a reduced effect size?). Can the authors provide further insight into the low compliance and the interpretation of the results, at least in the Discussion (and clarify better in the Abstract); even potentially citing previous literature that may clarify how effective CBT really is even with many drop outs.

3) It isn't clear how the patients and investigators were blinded; please better clarify since it doesn't seem blinding could be sufficiently possible since treatments are so different from each other that patients/investigators would know what treatment they received. A good place to describe this would be in the Randomisation and Blinding section of the Methods.

4) The AEs are provided in detail in the Supplementary Document. It would be helpful to provide some general overview/summary in the Results section around lines 320 to 322 to guide what types of AEs were observed and how many and if any dominant ones/types. Can the authors also provide if the AEs are related or not to treatment, since some seem to be not related at all; as well as if they resolved or not. If those details are not available, please indicate so. At least a sentence or two should summarize the overall safety/AE profiles of these different treatments since many times in studies with these different treatments, studies poorly capture and/or list the many AEs that can happen albeit mild so there is a biased view for patients that AEs rarely happen.

Additional suggestions:

-Line 158: is it meant to say "mild or worse cognitive impairment"

-Line 221: please provide more details in how the authors did the direct comparison of single vs combination treatments, i.e., that all the single treatments were pooled together and all the combo treatments were pooled together, and the randomization was done to ensure equivalent numbers. I figure it out after reading further along but would be helpful to state it clearly upfront for the reader.

-Figure 1: What is meant by 107 stratification group completed? Please clarify better.

-Line 426: It isn't clear how a placebo group would even be possible in this study, unless blinding is truly possible (see comment earlier). Please give an example of such a placebo arm in this section if possible.

Reviewer #2

(Remarks to the Author)

Line 84
Define UNITI

Line 86
CBT originally used a very specific orderly protocol in psychological counseling.
The first application of CBT for tinnitus was described by Sweetow (in Tyler, R. S. (Ed.). (2000). *Tinnitus Handbook*. San Diego, CA: Singular Publishing Group.
And by Henry and Wilson in their 2 books focused on Tinnitus
Unless the CBT for counseling follows a very formal protocol (such as by Henry and Wilson), the actual benefit depends mostly on the clinician.
The psychological literature has recently noted how CBT can be over-emphasized (see article cited in Tyler, R. S., Mohr, A. M. (2017). Is CBT for Tinnitus Overemphasized? *The Hearing Journal*, Journal Club 70(2):8,10, February 2017.)
CBT principals and exercises were included in *Tinnitus Activities Treatment*
Please describe specifically what you mean by CBT in this treatment.
Structured Counselling ??? there are many counseling procedures that are structured (for tinnitus too, such as *Tinnitus Activities Treatment*).

157

The sensitivity of the THI has been challenged. Please cite.
Tyler, R.S., Noble, W.G., Coelho, C. (2006). Considerations for the Design of Clinical Trials for Tinnitus. *Acta Oto-Laryngologica*, 126: 44-49.
Tyler, R.S., Oleson, J., Noble, W., Coelho, C., Ji, H. (2007). Clinical trials for tinnitus: Study populations, designs, measurement variables, and data analysis. *Progress in Brain Research*, 166: 499-509.
Also please describe how subjects scored the THI in your study? (Line 279 also).
189 if the structured counseling and sound therapy was self-administered, this would not apply to clinical services. Please be clear about this in the abstract, methodology and conclusions, as it would not apply to these services delivered in the clinic.

Line 193.

Is there a brief way you can describe the counseling and sound therapy without referring the reader to the "study protocol 16 ?? As it stands now, it will make it difficult to judge the validity of the study.

Line 210

There are many ways to measure numeric rating scales. Please describe these (labels, numbers, etc).

Line 228. CI – please spell out. Some readers might confuse with cochlear implants.

Line 237.... ITT.... please spell out for the reader.

Line 371... the lack of additional benefit of sound therapy is likely a result of self-administration without interaction over time with an audiologist. Please discuss this.

Line 385, the reader might want to know what an active comparator is.

Line 391. There are many counseling and sound therapy approaches administered by clinicians. You focus on TRT. TRT has set the field back many years. It uses "directive" not collaborative counseling. The mixing point is too high for most patients, and might increase tinnitus or increase hearing loss or interfere with speech perception.

It has been criticized in the literature by many (e.g. see the review ii

Tyler, R., Noble, W., Coelho, C., & Ji., H. (2012). Tinnitus Retraining Therapy: Mixing Point and Total Masking Are Equally Effective. *Ear Hear* 33(5):588–594

Of course, for marketing purposes people have modified TRT, making it more like strategies that existed before TRT (e.g. Tyler, R. S. & Babin, R. W. (1986). Tinnitus. In: C.W. Cummings, J.M. Fredrickson, L. Harker, C.J. Krause and D.E. Schuller (Eds.), *Otolaryngology Head and Neck Surgery* (3201-3217). St. Louis: C.V. Mosby Co.), but maintained the name TRT.

Bentler, R. A., & Tyler, R. S. (1987). Tinnitus management. *ASHA*, 29(5): 27-32.

Tyler, R. S., & Bentler, R. A. (1987). Tinnitus maskers and hearing aids for tinnitus. *Semin Hear*, 8(1): 49-61.

Tyler, R. S., Stouffer, J. L., & Schum, R. (1989). Audiological rehabilitation of the tinnitus client. *Journal of the Academy of Rehabilitative Audiology*, 22: 30-42.

Line 403. Indeed, the interpretation of the results are very limited and apply only to the specific applications provided. The high dropout rate in this study, questions the validity of the approaches used here. There there are numerous publications showing how counseling and sound therapy benefit patients, whether developed remotely or in person.

Line 423. Please cite some of the other treatments published, including TAT.

Line 431. I don't understand your suggestion of an active control condition. It will help the reader if you can be specific about what that would look like.

Line 443. I think this is widely known. It might be more helpful to suggest obstacles...cost... reimbursement.

I hope you find my comments helpful.

I would be happy to send reprints if needed.

Rich-tyler@uiowa.edu

Rich Tyler

The University of Iowa

Reviewer #3

(Remarks to the Author)

This is a large RCT involving 5 sites across 4 counties, and 10 treatment arms each using one or two treatments for tinnitus (of hearing aids, sound therapy, CBT, structured counselling). The primary comparison is between having one or two types of treatment for tinnitus. The results is a statistical but not clinically meaningful difference between the two groups. This is unsurprising given the extent of heterogeneity of treatments provided and compliance levels across the various groupings. Secondary and post hoc analyses suggest an advantage to offering combinations of treatments however, this cannot be formally concluded from the trial. The manuscript and reporting align with the published protocols and statistical analysis plan. There are some issues with the study design which will limit study quality and thus how much we can reliably interpret from the findings.

Randomisation is limited given those with indications for hearing aids, were only randomised across groups where hearing aids were received, and those who already had hearing aids were only randomised to no-hearing aid groups. Indication for hearing loss should be more clearly defined. Given the issue of hearing aid indicating or not, randomisation is not truly random, and any effects are likely diluted by the inclusion of existing hearing aid users. The trial should be describe as pseudo-randomised.

Please confirm explicitly which group is the control group (one or two treatments). This is I assume the single treatment group not clear but should be defined if a superiority trial. There is no true control group such as watchful waiting or no intervention. We know that tinnitus distress in general improves over time without intervention so if this small effect was factored into the analyses likely few comparisons would reach significance.

For completeness I suggest reporting the 48 week follow-up assessment data however incomplete. Was this the final visit as used in the tables?

Was there an equal/similar number of participants per site? If not what are the implications of this? Apologies if I missed this data.

There is some overlap in descriptions of efficacy/effectiveness, statistical and clinical significance. Given there is some suggestion of variability in treatment delivery and flexibility of treatment use, it sounds more a pragmatic effectiveness trial as opposed to efficacy. Was treatment fidelity assessed? Particularly for the self-directed elements how was this done. Structured Counselling is mislabelled. Counselling involved talking with a trained therapist, whereas here the term is used to describe some self-directed reading. Better to describe as self-directed reading or self-help.

The THI was not designed to measure outcome and with its limited scale lacks sensitivity to small but potentially important changes or differences. Hence it requires very large changes to suggest clinically meaningful change. This should be listed as a limitation. Lack of acknowledgement of the established Core Outcome Set for tinnitus intervention trials should also be

addressed. See Hall et al. (2018) The COMiT-ID study: developing core outcome domains sets for clinical trials of sound-, psychology-, and pharmacology-based interventions for chronic subjective tinnitus in adults. Trends in hearing, 22, 2331216518814384.

Why use multiple measures of tinnitus outcome (TFI, mini-TQ, 6 numerical scales) as secondary measures? If to capture different tinnitus outcomes, please specify.

Missing data doesn't appear to be completely missing at random – hearing aid recipients appeared the least compliant. Any reason for this?

Discussion should acknowledge the added benefit of hearing aids for hearing loss, and the limitation that this could be conflated with tinnitus benefit.

Reviewer #4

(Remarks to the Author)

The manuscript titled "Single versus Combination Treatment in Tinnitus: An International, Multicentre, Parallel-arm, Superiority, Randomised Controlled Trial" presents the results of a multinational research project aimed at assessing clinical effectiveness of a range of single treatments vs. selected combination treatments in the treatment of patients with tinnitus.

The study enrolled and randomized 461 patients with chronic tinnitus and mild or more severe tinnitus handicap across 5 sites (2x Germany, Belgium, Greece, Spain). Treatment groups consisted of single treatments with CBT, HA, SC, or ST, as well as their pairwise combinations. The primary outcome was the difference in change in THI from baseline to 12 weeks in single vs. combination treatments.

General Remarks:

The manuscript is well written and summarizes a complex research endeavor concisely, while at the same time providing a large range of information on the study conduct and proceedings as well as more detailed and sensitivity analyses results in appendices.

However, the research question, choice of study design, and lack of country-specific sensitivity analyses raise questions which should be addressed to provide a fully nuanced account of the study.

Firstly, the primary outcome strikes me as odd. Comparing (the mean of) all individual interventions with (the mean of) all combination treatments would suggest that the authors believe there is sufficient homogeneity in single and combination treatments per se as to account for a possible effect. In the discussion, the authors acknowledge that CBT has a record of effectiveness in tinnitus therapy, while ST performed poorly. If the authors believed there was heterogeneity in clinical effectiveness of single interventions, the more natural research question in my mind would be if each individual single therapy option benefits from a combination with another method or not. On the other hand, the current research question would yield valid insights into the real-world effectiveness of treatment on a population level if the distribution to single treatments were representative of that in actual patient care. However, as treatment groups were approximately balanced, it remains unclear what to learn from the aggregate effectiveness of single treatments. While I do not suggest that the authors change their original research question, it would help to understand the research question better if the apparent inconsistencies are addressed. In addition, presenting EMMs over time for single interventions in one graph would help to understand the variation in treatment effect by intervention.

Secondly, a multi-national study design was chosen, with sites across 4 countries in the EU. While the multi-national setup might be warranted by the funding grant for this research, I feel the current manuscript misses the opportunity to address potential variations in treatments and their effectiveness across the countries involved. While I acknowledge that a full investigation into location-specific effects might shift the focus of the manuscript and exceed capacity, I also believe the current results must be viewed in light of geographic location and differences in healthcare services. I would suggest to include a sensitivity analysis with "country * objective" as additional fixed effect and reporting EMMs by country. Also, the authors report the trial has been impacted by the COVID-19 pandemic and was terminated prematurely (but without a practical loss in statistical power). As countries were affected differently across Europe, it would help to see the group allocation not only broken down by treatment group, but also by country to understand if recruitment and/or drop-outs were affected differently by country.

Thirdly, approximately 20% of MFI data was reported as missing. The authors assume data was MAR, and describe the procedures for assessing missing data in the SAP. While an assessment of the missingness mechanism always remains speculative, and some drop-out data hints at treatments being perceived as either not effective or impractical, it would help to put results into context if the authors report results under MNAR conditions.

To conclude, the current manuscript provides an excellent basis for expanding results and making the assumptions and conclusions more traceable.

Version 1:

Reviewer comments:

Reviewer #1

(Remarks to the Author)

The authors have sufficiently addressed my concerns; exciting and impactful work!

Reviewer #2

(Remarks to the Author)

2nd review UNITI

157 and 166

Please state how subjects were asked to score the THI.

Widely used does not mean valid.

The THI actually uses a 3-label category scale. Your numeric total scores, suggest you did not actually use the THI.

Reviewer #3

(Remarks to the Author)

Thank you for the response to reviewer comments and revisions made. I do not have further to add but to reiterate,

The trial is not truly random but I appreciate a stratified randomization procedure was used.

I disagree with the authors use of the term Counselling in place of self-help/self-management.

I consider the report incomplete without the 48-week data. Will this data be made available, e.g. for critique in systematic reviews – it is not clear from the data availability statement. Also would rephrase your rationale for not reporting it - all research data is collected from volunteers.

I am unconvinced on the choice of THI as the primary outcome. Your strongest argument is that it conforms with the TFI and TQ – so why not use the TFI then which by design has greater sensitivity to smaller changes with its 11-point item scales versus the 3-point scales on the THI.

Reviewer #4

(Remarks to the Author)

Thank you for the additional information provided in response to my comments and questions. However, I would still recommend to implement the following analyses to challenge your assumptions and strengthen the conclusions you draw from the results.

1) Although protocols for treatments have been harmonized across countries, differences in individual interventions' effectiveness might still exist due to variations in implementation and acceptance of the patient population for the individual treatments. A country-specific analysis of possible variations in treatment and their effectiveness is the prerequisite before drawing conclusions from the pooled data. I strongly recommend conducting this sensitivity analysis.

2) Although MICE is a widely adopted imputation approach for data believed to be MAR, the reasons for drop-out stated in tables S7 and S8 suggest that in many cases, drop-outs occurred due to dissatisfaction with an intervention or no perceived benefit and worsening tinnitus symptoms. This suggests that some portion of data is MNAR, which needs to be addressed by an appropriate imputation mechanism. I strongly suggest the authors conduct a sensitivity analysis using a reference-based imputation approach (e.g. <https://journals.sagepub.com/doi/10.1177/1536867X1601600211>) to challenge their assumptions.

Thank you again for this interesting work and your effort in addressing all comments and questions this far!

Version 2:

Reviewer comments:

Reviewer #3

(Remarks to the Author)

Thank you for your further responses. I am still concerned that much of the study relied on pragmatic decision making and opinion, e.g., use of THI as a primary outcome measure.

I absolutely disagree there are no generally agreed definitions of counselling. For such you should look to the relevant societies rather than falling on non-expert perspectives, e.g. the American Counseling Association clearly defines

counselling as "a learning-oriented process, carried on in a simple, one-to-one social environment, in which a counselor, professionally competent in relevant psychological skills and knowledge, seeks to assist the client, by methods appropriate to the latter's needs and within the context of the total personnel program, to learn more about himself and to accept himself, to learn how to put such understanding into effect in relation to more clearly perceived, realistically defined goals to the end that the client may become a happier and more productive member of his society."

British Psychological Society definition: Counselling and psychotherapy are forms of 'talking therapy' for personal issues, such as stress, worry, anxiety or depression. Counselling may be briefer and provides a listening space, while psychotherapy may be longer and look more closely at emotional or behavioural difficulties. They involve sharing your problems with your therapist or counsellor in a confidential setting.

A textbook definition of counselling is a contracted meeting between a client and a counsellor. The meeting happens at a set time, in an agreed place, for the sole benefit of the client.

Counselling happens at a specified time and at a specific place, and the sole focus of the meeting is to benefit the client.

Reviewer #4

(Remarks to the Author)

Thank you for including the additional analyses and discussing results more broadly! I have no further requests and thank the authors for their contribution.

Response to the reviewers of the manuscript entitled:

**Single versus Combination Treatment in Tinnitus: An International, Multicentre,
Parallel-arm, Superiority, Randomised Controlled Trial**

Regensburg, December 17, 2024

Dear reviewers,

thank you very much for the positive and constructive feedback as well as for appreciating our work. Please find our answers in a point-by-point format below each of your comments and raised points. Changes in the revised version of the manuscript are highlighted in red and appended to each response together with the line number.

Hopefully, the changes to our manuscript are appropriate and you will consider it for publication.

Yours sincerely,

Stefan Schoisswohl

(on behalf of all authors)

Reviewer #1 (Remarks to the Author):

The authors have performed a much needed and challenging clinical study in the tinnitus field that assessed several routine treatments either alone or in combination. It is not uncommon for many tinnitus patients to try multiple treatments for tinnitus; yet the efficacy of one or a series of treatments is not well understood. Although quite challenging to assess so many different treatments and combinations, the authors should be commended for pursuing an initial study to try to better understand what is pursued in real world clinics and performing a quite large study to have enough power to test a few comparisons. Overall, the paper is well written and clear, and rigorously done with pre-specified protocol/SAPS that were published beforehand. They were able to show that combo treatments can be effective but not necessarily better than single treatments; instead, more to offer alternative options in which the better performing treatment for a given patient is mainly what dictates their overall outcomes. That finding is still clinically impactful because it further supports that patients should still be provided different options and combinations since they have a chance of benefiting sufficiently based on one that is most effective for them. Furthermore, ST has proven to be minimally effective but can still be useful combined with other approaches, while HAs had the largest effect size (see comment below but not fully convincing if CBT should also be considered as also having large effect size). There are a few suggestions provided below to potentially further help the clarity of the paper and its interpretation.

>RESPONSE: Thank you very much for your positive feedback, your careful review, and constructive comments in order to improve the manuscript.

Four general concerns/considerations:

1) The study deliberately stratified patients into four equally sized strata based on THI total score (cutoff of 48) and hearing aid indication (yes/no). It was not clearly justified how the

authors came up with these strata, so it would be helpful to better clarify why these strata were selected. More importantly, if so important to stratify based on those groups, then those results should be included in the main paper and some key interpretations/conclusions should be presented in the paper about them (only saw 2 tables lightly mentioned in the Supplementary document). Please provide additional plots or results in the main paper to interpret those findings or justify why they are no longer critical for the original design/expectations of the study.

>RESPONSE: The decision to use a cut-off of 48 in the THI total score for stratifying patients into low and high tinnitus-related handicap groups was based on the median THI score calculated from historical data of N = 837 patients from the clinical site in Regensburg and further discussed in a round of clinical experts in the field.

The cut-offs for HA indication as can be seen in Table S2 in the Supplementary Material were defined by a group of experts in the field of ENT, audiology and hearing aids.

We added this information to the methods part of the manuscript under the section “Randomisation and blinding”.

Line number 171 – 175:

“Criterion for low and high tinnitus-related handicap was defined based on historical data obtained from 837 patients at the clinical site in Regensburg with a median THI score of 48. Hearing aid indication criteria were specified by a group of international experts in the fields of audiology and otolaryngology (see **Table S3**).”

We stratified into low and high tinnitus handicap in order to ensure that tinnitus severity at baseline was in a similar range for all treatment arms. The stratification in patients with and without hearing aid indication was necessary, as the treatment option “hearing aid” was only appropriate for patients with a hearing loss with hearing aid indication. We make this now clearer for the reader and added the following in the text:

Line number 177 – 182:

“Patients from the two strata without hearing aid indication were not randomised in treatment groups that comprised HA treatment. **The stratification according to tinnitus-related handicap was performed to ensure an equal representation of patients with high and low tinnitus distress in different treatment arms and thus avoid potential misinterpretations of our findings due to large differences in baseline tinnitus severity across treatment arms.**”

From our understanding, there is no need to further present the “findings” of the stratification process, since this process rather represents an approach to improve the interpretability and transferability of potential findings. We hope that this proceeding is now clear.

2) CBT has been referred to in the paper (e.g., Discussion) as the most established treatment and in this current study there is an interpretation that it alone performed well (at least compared to ST). However, from Figure 1, it also had the largest drop-out rate. It isn't clear how multiple imputation or the current analyses can support the efficacy of CBT when so many have dropped out specifically for CBT (e.g., are those who dropped out non-responders or would ultimately cause a reduced effect size?). Can the authors provide further insight into the low compliance and the interpretation of the results, at least in the Discussion (and clarify better in the Abstract); even potentially citing previous literature that may clarify how effective CBT really is even with many dropouts.

>RESPONSE: We thank the reviewer for pointing this out. Indeed, CBT - as well as its combination with other treatment types - exhibits the highest dropout rates. We added the following part to our discussion and also added the term dropouts to the legend of Figure1:

Line number 478 – 490:

“CBT treatment arms, which require a high level of commitment with several on-site visits, demonstrated the highest proportion of dropouts in our trial, which potentially limits the

interpretability and robustness of our CBT findings, as non-responders may be overrepresented among dropouts. In another recent study, in which CBT was compared with Neurofeedback, the CBT dropout rate was in a similar high range like in our study.⁵³ There is a large body of evidence in the literature that CBT is effective in the treatment of tinnitus (for an overview see the Cochrane review by Fuller et al., 2020),³⁹ and has been recommended in European guidelines for the management of tinnitus.⁸ However, all studies investigating CBT alone might be susceptible to a selection bias, as only patients with motivation for CBT would have been enrolled. The relatively high dropout rate of CBT in studies comparing various treatment options reflects the clinical experience of the real-world situation where a relevant subgroup of patients is not willing to undergo CBT. Detailed information on dropout reasons per treatment arm are listed in **Tables S6 – S9.**”

Line number 306:

“Quantity and reasons for trial exclusion during eligibility assessments and trial discontinuation/dropouts can be seen from **Tables S5 – S9.**”

3) It isn't clear how the patients and investigators were blinded; please better clarify since it doesn't seem blinding could be sufficiently possible since treatments are so different from each other that patients/investigators would know what treatment they received. A good place to describe this would be in the Randomisation and Blinding section of the Methods.

>RESPONSE: It seems that there is a misunderstanding with respect to blinding. Patients and assessors could not be blinded to the respective treatment group, as this was impossible due to different types of treatment. However, the statistical analysis team was blinded until the analyses were finished. We tried to highlight this in the section “Randomisation and blinding”:

Line number 185 – 188:

“Treatment codes were used to ensure blindness of the statistical analysis team to the type of treatment patients received. Unblinding was conducted after analyses completion. Patients and investigators/assessors were not blinded. See study protocol and statistical analysis plan for more detailed information.^{17,18}”

4) The AEs are provided in detail in the Supplementary Document. It would be helpful to provide some general overview/summary in the Results section around lines 320 to 322 to guide what types of AEs were observed and how many and if any dominant ones/types. Can the authors also provide if the AEs are related or not to treatment, since some seem to be not related at all; as well as if they resolved or not. If those details are not available, please indicate so. At least a sentence or two should summarize the overall safety/AE profiles of these different treatments since many times in studies with these different treatments, studies poorly capture and/or list the many AEs that can happen albeit mild so there is a biased view for patients that AEs rarely happen.

>RESPONSE: We thank the reviewer for bringing this up. We updated this section and added a sentence about the general safety of the treatments used.

Line number 359 – 371:

“No SAE was evident in any patient. AEs appeared in 49 (21.3%) participants in single treatment groups, and in 49 (21.2%) participants in combination treatment groups. The most relevant AEs reported by patients were worsening of the tinnitus percept (6); worsening of their psychological health (3); sleep problems (2); pain in the ear when wearing the hearing aid (1), ear infection (1), inflammation of the ear (1), dizziness (1), and mild transient hearing loss (1). Worsening of tinnitus symptoms is a relative common side-effect in tinnitus studies, as patients are focussing their attention more intently on their tinnitus to evaluate potential changes in tinnitus characteristics. Given the absence of any SAE and the low number of adverse reactions associated potentially with the various treatments, the present intervention types can be

considered as safe. As AEs were rather rare and not severe, we abstained from analysing the strength of the relationship with treatment interventions and from documenting the time course of the reported AEs. A full listing of all AEs per treatment arm is provided in Table S11. Information on treatment adherence is given in Figure S1.”

Additional suggestions:

-Line 158: is it meant to say "mild or worse cognitive impairment"

>RESPONSE: Thank you for raising this. Yes, we mean “mild or worse” cognitive impairment. We updated this part:

Line number 159-161:

“Exclusion criteria were: presence of a mild or worse cognitive impairment according to the Montreal Cognitive Assessment²⁰ (MoCa; score ≤ 22);...”

-Line 221: please provide more details in how the authors did the direct comparison of single vs combination treatments, i.e., that all the single treatments were pooled together and all the combo treatments were pooled together, and the randomization was done to ensure equivalent numbers. I figure it out after reading further along but would be helpful to state it clearly upfront for the reader.

>RESPONSE: Thank you for this point. We added a reference to the SAP, where detailed information on treatment arm pooling for the contrasts of interest can be found.

Line number 281 – 282:

“Detailed information on which treatment arms were pooled for which type of comparison can be found in the SAP.¹⁸”

Figure 1: What is mean by 107 stratification group completed? Please clarify better.

>RESPONSE: As we aimed for equally sized stratification groups, each center had 25 randomization slots per stratification group. Once a stratification group e.g., patients with high THI score and no hearing aid indication was full (reached n=25), a patient with this specific profile could not be included in the trial. We tried to explain this in more detail as a note to Table S5 in the Supplementary:

“Note: We aimed for four equally sized stratification groups of 25 patients per clinical site. 107 patients could not be included in the trial since the respective stratification group (e.g., high tinnitus-related handicap and no hearing aid indication) was already full.”

-Line 426: It isn't clear how a placebo group would even be possible in this study, unless blinding is truly possible (see comment earlier). Please give an example of such a placebo arm in this section if possible.

>RESPONSE: We agree with the reviewer that a double-blinded placebo treatment arm is not be feasible in the context of the present trial with many different treatments and combinations. Therefore, we changed the term “placebo” to “control” as an unblinded control group without a specific therapeutic intervention would have been theoretically feasible for a study of this type.

Line number 508-510:

“A **control** group was not included in this trial, as the answer to the main question (comparison of single and combined treatment) did not require a control **group**. Nevertheless, a **control** group may have been helpful as an anchor for comparison with the ten treatment arms.”

Reviewer #2 (Remarks to the Author):

>RESPONSE: Thank you very much for your careful review, constructive comments and the suggestions for further references in order to improve the manuscript.

Line 84

Define UNITI

>RESPONSE: We now defined UNITI in the abstract of the manuscript.

Line number 84 - 85:

“...sites across the EU as part of the interdisciplinary collaborative UNITI project
(Unification of Treatments and Interventions for Tinnitus Patients).”

Line 86

CBT originally used a very specific orderly protocol in psychological counseling.

The first application of CBT for tinnitus was described by Sweetow (in Tyler, R. S. (Ed).

(2000). Tinnitus Handbook. San Diego, CA: Singular Publishing Group.

And by Henry and Wilson in their 2 books focused on Tinnitus

Unless the CBT for counseling follows a very formal protocol (such as by Henry and

Wilson), the actual benefit depends mostly on the clinician.

The psychological literature has recently noted how CBT can be over-emphasized (see

article sited in Tyler, R. S., Mohr, A. M. (2017). Is CBT for Tinnitus Overemphasized? The

Hearing Journal, Journal Club 70(2):8,10, February 2017.)

CBT principals and exercises were included in Tinnitus Activities Treatment

Please describe specifically what you mean by CBT in this treatment.

***Structured Counselling ??? there are many counseling proceducrees that are structured (for
tinnitus too, such as Tinnitus Activities Treatment).***

>RESPONSE: We thank the reviewer for pointing out these aspects. We added some more details about CBT and structured counselling. As the space is limited, we referenced to additional literature with more details about the interventions applied in the project.

Line number 191 – 210:

“Single and combination treatments were applied over a 12-week treatment phase. All treatment procedures were designed by dedicated experts in their respective fields (see Table S3 for expert team per treatment type) and described in detail in the study protocol.¹⁷ To ensure consistency with respect to treatment and assessment implementation across clinical sites, workshops were held, and Standard Operation Procedure documents were created. Two of the four treatment types were unguided, app-based therapies, minimising potential differences. HA fitting was standardised, and CBT was co-developed specifically for this trial.

CBT was based on the concept of fear-avoidance using exposure therapy.^{22,23} The exposure exercises were delivered by trained psychologists or psychotherapists in weekly face-to-face group sessions (1.5-2 hours weekly; 12 weeks; group size: six to eight participants). For HA treatment, behind-the-ear hearing instruments (Type Signia Pure 312 7X; Sivantos Pte. Ltd., Singapore, Republic of Singapore/ WSAudiology, Lynge, Denmark) were fitted bilaterally with all noise-related signal processing deactivated by audiologists or HA acousticians according to the National Acoustic Laboratories-Non-Linear 2 generic amplification proceeding.²⁴ SC and ST were self-administered on a daily basis via a dedicated UNITI mobile application, which was available for Android and iOS devices as well as free of charge.²⁵ SC was oriented on recent European guidelines for tinnitus management⁸ and consisted of 12 chapters featuring structured patient education (e.g., facts about tinnitus, brain and sound perception; myths and misconceptions about tinnitus; diagnosis of tinnitus; special types of tinnitus; therapeutic approaches; psychological and behavioural aspects) and tips on how to handle tinnitus distress.”

The sensitivity of the THI has been challenged. Please cite.

Tyler, R.S., Noble, W.G., Coelho, C. (2006). Considerations for the Design of Clinical Trials for Tinnitus. Acta Oto-Laryngologica, 126: 44-49.

Tyler, R.S., Oleson, J., Noble, W., Coelho, C., Ji, H. (2007). Clinical trials for tinnitus: Study populations, designs, measurement variables, and data analysis. Progress in Brain Research,

>RESPONSE: Indeed, there is some critique on the sensitivity of the THI. We added this point to the section on outcome measures and cited the two papers that you suggested.

Line number 228 - 235:

“The primary outcome **between single and combination treatment was** the difference in change from baseline to final visit (after 12 weeks of treatment) in the Tinnitus Handicap Inventory (THI),²⁰ which consists of 25 items to quantify tinnitus handicap (total scores range: 0-100). Changes from baseline to interim visit, and follow-up were examined in secondary analyses as well. **Despite some critique on its sensitivity,^{27,28} the THI was chosen as the primary outcome measure, since i) it is the most widely used clinical instrument in research,^{29,30} and ii) there is high evidence of a conformity between the THI, the Tinnitus Functional Index (TFI)³¹, and the Tinnitus Questionnaire (TQ).³²”**

166: 499-509.

Also please describe how subjects scored the THI in your study? (Line 279 also).

>RESPONSE: Thank you for this point. We added the information to the manuscript.

Line number 250-252:

“**Questionnaires were filled out by the patients using a graphical interface of the UNITI database.¹⁷ Patients could also opt for paper-pencil versions, and data was subsequently entered into the UNITI database by the local study team.”**

189 if the structured counseling and sound therapy was self-administered, this would not apply to clinical services. Please be clear about this in the abstract, methodology and conclusions, as it would not apply to these services delivered in the clinic.

>RESPONSE: We thank the reviewer for this comment. However, even if available on an app, structured counselling and sound therapy can be proposed by the clinic, whose role is to follow-up with the patient for improvements. As an example, Lenire's bimodal therapy has to be done at home by the patient, not at the clinic – here the clinic is responsible for following-up and ensuring treatment efficacy and safety.

We include additional information to the abstract and also to the relevant sections:

Line number 86 – 89:

“Cognitive-behavioural therapy, hearing aids, structured counselling (app-based), and sound therapy (app-based) were administered either alone or as a combination of two treatments resulting in ten treatment arms.”

Line number 213 – 215:

“There are many different SC and ST approaches administered by clinicians. For clarity, we want to mention that our app-based approach did not follow the Tinnitus Retraining Therapy protocol.”

Line number 466 – 471:

“Any interpretation of our findings should keep in mind, that we investigated specific applications of CBT, HA, ST, and SC. Potential reasons for the low efficacy of ST and SC in the present trial might include its self-administration, the limited interaction with a clinical specialist and/or the absence of specific instructions (stimulus, loudness, duration etc.). Thus, our conclusions on ST and SC might not be directly applied to a traditional clinical setting, where patients are not necessarily followed-up.”

Line 193.

Is there a brief way you can describe the counseling and sound therapy without referring the reader to the “study protocol 16 ?? As it stands now, it will make it difficult to judge the validity of the study.

>RESPONSE: Thank you for this suggestion. We added the relevant information for SC and ST.

Line number 204 – 215:

“SC and ST were self-administered on a daily basis via a dedicated **UNITI** mobile application, which was available for Android and iOS devices as well as free of charge.²⁵ SC was oriented on the recent European guidelines for tinnitus management⁸ and consisted of 12 chapters featuring structured patient education (e.g., facts about tinnitus, brain and sound perception; myths and misconceptions about tinnitus; diagnosis of tinnitus; special types of tinnitus; therapeutic approaches; psychological and behavioural aspects) and tips on how to handle tinnitus distress. ST included 64 different artificial and naturalistic sounds with various state of the art modulation or filter techniques. Loudness and length of the sounds was adjustable by the patients. There are many different SC and ST approaches administered by clinicians. For clarity, we want to mention that our app-based approach did not follow the Tinnitus Retraining Therapy protocol.”

Line 210

There are many ways to measure numeric rating scales. Please describe these (labels, numbers, etc).

>RESPONSE: You are right, there are many ways to measure NRS. We included the information how the scale was anchored in our manuscript.

Line number 238 – 243:

“... as well as numeric rating scales (NRS; 0 - 10) for tinnitus impairment (0 - not a problem; 10 - very big problem), tinnitus loudness (0 - not at all loud; 10 - extremely loud), tinnitus-related discomfort (0 - no discomfort; 10 - severe discomfort), annoyance (0 - not at all annoying; 10 - extremely annoying), unpleasantness (0 - not at all unpleasant; 10 - extremely unpleasant), and ability to ignore the tinnitus (0 - very easy to ignore; 10 - impossible to ignore).³⁶”

Line 228. CI – please spell out. Some readers might confuse with cochlear implants.

>RESPONSE: Thank you for spotting this mistake. We have corrected this in the revised version.

Line number 95 - 96:

“Least-squares mean changes from baseline to week 12 were -11.7 for single treatment (95% confidence interval [CI], -14.4 to -9.0) and...”

Line 237.... ITT.... please spell out for the reader.

>RESPONSE: We already spelled out ITT when we first mentioned the term in the text (section *Statistical Analysis*). However, we realized, that we have many abbreviations in our text. Therefore, we spelled out all abbreviations in the discussion section once again in order to ensure a better reading flow. Thank you very much for bringing this up.

Line number 263 – 264:

“The statistical analysis was performed in the intention-to-treat (ITT) population of N = 461, including all randomised participants, regardless of...”

Line number 412 – 414:

“...of established tinnitus treatments (cognitive-behavioural therapy (CBT), hearing aids (HA), structured counselling (ST), and sound therapy (ST)) applied either...”

Line number 476 – 477:

“...was low (see **Figure S1** and **Table S35**) and dropout rates were high in our trial (**per-protocol (PP)** sample of 185 patients).”

Line number 498:

“...incongruency between **intention-to-treat (ITT)** and PP analysis...”

Line 371... the lack of additional benefit of sound therapy is likely a result of self-administration without interaction over time with an audiologist. Please discuss this.

>RESPONSE: This is a good point for the discussion – thank you. We added his part to our manuscript:

Line number 467 – 471:

“**Potential reasons for the low efficacy of ST and SC in the present trial might include its self-administration, the limited interaction with a clinical specialist and/or the absence of specific instructions (stimulus, loudness, duration etc.). Thus, our conclusions on ST and SC might not be directly applied to a traditional clinical setting, where patients are not necessarily followed-up.**”

Line 385, the reader might want to know what an active comparator is.

>RESPONSE: Thank you for this comment. We changed "active comparator" to "active control" - consistent to the original article that is cited here.

Line number 436 – 437:

“The weak clinical **effectiveness** of sound treatment alone is in line with previous work where sound treatment was used as an active **control**.⁴²”

Line 391. There are many counseling and sound therapy approaches administered by clinicians. You focus on TRT.

TRT has set the field back many years. It uses “directive” not collaborative counseling. The

mixing point is too high for most patients and might increase tinnitus or increase hearing loss or interfere with speech perception.

It has been criticized in the literature by many (e.g. see the review ii

Tyler, R., Noble, W., Coelho, C., & Ji, H. (2012). Tinnitus Retraining Therapy: Mixing Point and Total Masking Are Equally Effective. Ear Hear 33(5):588–594

Of course, for marketing purposes people have modified TRT, making it more like strategies that existed before TRT (e.g.

Tyler, R. S. & Babin, R. W. (1986). Tinnitus. In: C.W. Cummings, J.M. Fredrickson, L. Harker, C.J. Krause and D.E. Schuller (Eds.), Otolaryngology Head and Neck Surgery (3201-3217). St. Louis: C.V. Mosby Co.), but maintained the name TRT.

Bentler, R. A., & Tyler, R. S. (1987). Tinnitus management. ASHA, 29(5): 27 32.

Tyler, R. S., & Bentler, R. A. (1987). Tinnitus maskers and hearing aids for tinnitus. Semin Hear, 8(1): 49 61.

Tyler, R. S., Stouffer, J. L., & Schum, R. (1989). Audiological rehabilitation of the tinnitus client. Journal of the Academy of Rehabilitative Audiology, 22: 30 42.

>RESPONSE: We agree with the reviewer. There are many counselling and sound therapy approaches. In our study, we used approaches that are different from TRT. We clarified this in the manuscript and added the following sentence in the description of the procedures.

Line number 213 – 215:

“There are many different SC and ST approaches administered by clinicians. For clarity, we want to mention that our app-based approach did not follow the Tinnitus Retraining Therapy protocol.”

Line 403. Indeed, the interpretation of the results are very limited and apply only to the specific applications provided. The high dropout rate in this study, questions the validity of the approaches used here. There are numerous publications showing how counseling and sound therapy benefit patients, whether developed remotely or in person.

>RESPONSE: We fully agree. Motivated by your comment, we made this point even stronger in the discussion section:

Line number 463 – 471:

“It should also be considered that we worked with a selected set of four tinnitus treatments and combinations of **only** two treatment types. Thus, it remains unknown, whether the combination of other treatment sets or combinations of three or more treatment types would lead to additional treatment benefits. **Any interpretation of our findings should keep in mind, that we investigated specific applications of CBT, HA, ST, and SC. Potential reasons for the low efficacy of ST and SC in the present trial might include its self-administration, the limited interaction with a clinical specialist and/or the absence of specific instructions (stimulus, loudness, duration etc.). Thus, our conclusions on ST and SC might not be directly applied to a traditional clinical setting, where patients are not necessarily followed-up.**”

Line 423. Please cite some of the other treatments published, including TAT.

>RESPONSE: We now include TAT as an additional example of combined treatment in the introduction.

Line number 125 – 128:

“However, studies on the **effectiveness** of combining clinical interventions are scarce.^{9–11} **Prominent examples** of combining different treatment types **are** represented by the combination of acoustic therapy with directive counselling **as implemented in the Tinnitus Activities Treatment¹² or the Tinnitus Retraining Therapy.¹³**”

Line 431. I don't understand your suggestion of an active control condition. It will help the reader if you can be specific about what that would look like.

>RESPONSE: Please see the sentence below about how this would help in the context of the present study.

Line number 512 – 516:

“Further, our data demonstrates low **effectiveness** of ST as a single treatment, supporting its use as an active control condition in randomised controlled trials.^{45,46} Thus, the two treatment arms CBT and ST can be considered as reliable reference anchors for the interpretation of the results of the other 8 investigated treatment arms.”

Line 443. I think this is widely known. It might be more helpful to suggest obstacles....cost... reimbursement.

>RESPONSE: Even if effects of SC and HA have been assumed based on clinical experience, there is only limited evidence for their effectiveness. According to our knowledge this is the first study in which the effectiveness of SC, HA and CBT were directly compared against each other.

Concerning obstacles, we discuss the low treatment adherence (line number: 503-507):

“Our results indicate that there is a high need for further research to better understand the clinical **benefits** of combination treatment; to get more profound insights behind the reasons for low treatment adherence; and in approaches to increase treatment adherence in daily clinical practice, such as the implementation of behavioral change techniques or more extensive patient education.”

We agree that there are further obstacles that impede widespread implementation of our findings in clinical routine. However, these obstacles are complex, as they depend on availability, national health systems (e.g. public vs private) or national reimbursement regulations. The identification of these obstacles was not the focus of our study, but we agree, that they should be addressed by further research. Some of our team members have been at the management of the European guidelines. This could be a first step towards a harmonised consensus and evidence-based establishment of recommendations, prior their use in clinical

routines.

Reviewer #3 (Remarks to the Author):

This is a large RCT involving 5 sites across 4 counties, and 10 treatment arms each using one or two treatments for tinnitus (of hearing aids, sound therapy, CBT, structured counselling). The primary comparison is between having one or two types of treatment for tinnitus. The results is a statistical but not clinically meaningful difference between the two groups. This is unsurprising given the extent of heterogeneity of treatments provided and compliance levels across the various groupings. Secondary and post hoc analyses suggest an advantage to offering combinations of treatments however, this cannot be formally concluded from the trial. The manuscript and reporting align with the published protocols and statistical analysis plan. There are some issues with the study design which will limit study quality and thus how much we can reliably interpret from the findings.

>RESPONSE: Thank you very much for your positive feedback, your careful review and constructive comments to improve the manuscript. Here below, we answer your comments and open questions.

Randomisation is limited given those with indications for hearing aids, were only randomised across groups where hearing aids were received, and those who already had hearing aids were only randomised to no-hearing aid groups. Indication for hearing loss should be more clearly defined. Given the issue of hearing aid indicating or not, randomisation is not truly random, and any effects are likely diluted by the inclusion of existing hearing aid users. The trial should be describe as pseudo-randomised.

>RESPONSE: We diligently disagree with the comments of the reviewer. We randomised patients without a hearing aid indication only to treatment arms without hearing aids (6 arms), but, and this is very important, patients with a hearing aid indication were randomly allocated to all treatment arms (10 arms), those with hearing aids (4 arms) and those without hearing aids (6 arms). Please see the randomization schedule in our published study protocol

(Schoisswohl et al., 2021, DOI: 10.1186/s13063-021-05835-z) and in our published statistical analysis plan (Simoes et al., 2023, DOI: 10.1186/s13063-023-07303-2) as well as the section “randomization and blinding” in our manuscript.

Line number 175 – 182:

“Participants were then randomised to one of ten treatment arms comprised of single (CBT, HA, SC, ST) and combination interventions (CBT+HA, CBT+SC, CBT+ST, HA+SC, HA+ST, SC+ST) **under consideration of the stratification group**. Patients from the two strata without hearing aid indication were not randomised in treatment groups that comprised HA treatment. **The stratification according to tinnitus-related handicap was performed to ensure an equal representation of patients with high and low tinnitus distress in different treatment arms and thus avoid potential misinterpretations of our findings due to large differences in baseline tinnitus severity across treatment arms.**”

As a consequence, we will maintain the wording “randomisation” and not use “pseudo-randomised”.

The Criteria for hearing aid indication can be found in the Supplementary Material Table S2.

We now include additional information on how we came up with these criteria.

Line number 173 – 175:

“**Hearing aid indication criteria were specified by a group of international experts in the fields of audiology and otolaryngology (see Table S3).**”

We agree that the inclusion of patients using hearing aids may represent a potential confounder. However, we aimed at a representative sample of tinnitus patients and decided not to exclude patients using hearing aids. To minimize potential confounding influences, we included hearing aid indication as a cofactor in our statistical models. We made this clearer in the revised version of the manuscript:

Line number 286 - 288:

“The models were adjusted for the following covariates: age, sex, educational attainment, hearing aid indication, and PHQ-9 baseline scores.¹⁸”

Please confirm explicitly which group is the control group (one or two treatments). This is I assume the single treatment group not clear but should be defined if a superiority trial.

>RESPONSE: Thank you for your suggestion. We added the following parts in the introduction and methods part of the manuscript:

Line number 129 – 130:

“The primary objective of the current trial was to **investigate if combination treatments are more effective than single treatments for** patients with chronic tinnitus.”

Line number 276 – 277:

“The analysis of the primary objective was performed in the ITT population to test the **effectiveness** of combination treatments against single treatments (**control group**).”

There is no true control group such as watchful waiting or no intervention. We know that tinnitus distress in general improves over time without intervention so if this small effect was factored into the analyses likely few comparisons would reach significance.

>Indeed, this a limitation of the study and we tried to address this in the discussion section – see below. However, since the focus of the trial was to compare single and combination treatments, plus the interventions used were already proven to be effective, a control group such as a waiting list control group was not included in the present trial.

Line number 508 – 510:

“A **control** group was not included in this trial, as the answer to the main question (comparison of single and combined treatment) did not require a control **group**. Nevertheless,

a **control** group may have been helpful as an anchor for comparison with the ten treatment arms.”

For completeness I suggest reporting the 48 week follow-up assessment data however incomplete. Was this the final visit as used in the tables?

>RESPONSE: It was an informed decision from our side to not include the 48-week follow-up in our manuscript and the main analyses (only the 36-week follow-up), since the 48-week follow-up was on a voluntary basis (see study protocol) and not every clinical site performed it. Thus, the data of the 48-week follow-up should only be used for exploratory analyses. Therefore, we preferred not to include data from the additional facultative follow-up at 48 weeks because of many missings and the lack of standardized instructions for data collection.

Was there an equal/similar number of participants per site? If not what are the implications of this? Apologies if I missed this data.

>RESPONSE: Our goal was to recruit 100 patients per clinical site – please see the published study protocol and the statistical analysis plan. However, due to delayed recruitment and inclusion processes because of the COVID pandemic on one side and a limited project duration on the other side, we were forced to close the trial even before the targeted sample size was reached. We already stated this in the results section, and we now added the numbers on how many patients were included per country:

Line number 316-317:

“... with N = 461 included and treated patients, in order to keep to the schedule of our funding period (**Granada: 89, Athens: 99, Leuven: 74, Regensburg: 100, Berlin: 99**).”

There is some overlap in descriptions of efficacy/effectiveness, statistical and clinical significance.

>RESPONSE: Thank you for bringing this up. We clarified the terms throughout the whole manuscript.

Given there is some suggestion of variability in treatment delivery and flexibility of treatment use, it sounds more a pragmatic effectiveness trial as opposed to efficacy. Was treatment fidelity assessed? Particularly for the self-directed elements how was this done.

>RESPONSE: We agree that the trial was primarily an effectiveness trial and we clarified this throughout the manuscript.

We assessed compliance to the treatment protocol per intervention. The compliance definitions were previously published in the statistical analysis plan. We added the description how we assessed treatment compliance per intervention type.

Line number 216 – 219:

“Treatment compliance was assessed via participation in CBT treatment sessions (≥ 6 CBT sessions; including the first two), usage log files for HAs (average use of ≥ 4 hours/day) and App-use logfiles for SC (completion of the first six chapters) and ST (using each of the four sound stimuli categories once)¹⁸”

Structured Counselling is mislabelled. Counselling involved talking with a trained therapist, whereas here the term is used to describe some self-directed reading. Better to describe as self-directed reading or self-help.

>RESPONSE: You are right, counselling typically involves talking to a trained therapist.

However, there is increasing evidence that the counselling content can also be conveyed via digital tools and is also as effective as being delivered by a trained therapist.

We consider the term appropriate, as we are clearly stating in the manuscript that SC has been self-administered via a mobile app. The creation of the SC protocol followed the recent European Tinnitus Guidelines and was designed and labelled by experts in the field of psychological interventions in tinnitus, and an international consortium (UNITI-consortium) agreed on this labelling.

We clarified throughout the manuscript that SC was app-based and we stated in the discussion that the way, how the various treatments were administered, should be considered in the interpretation of the results.:

Line number 86 – 87:

“..., structured counselling (app-based), and sound therapy (app-based) were administered ...“

Line number 204 – 215:

“SC and ST were self-administered on a daily basis via a dedicated UNITI mobile application, which was available for Android and iOS devices as well as free of charge.²⁵ SC was oriented on the recent European guidelines for tinnitus management⁸ and consisted of 12 chapters featuring structured patient education (e.g., facts about tinnitus, brain and sound perception; myths and misconceptions about tinnitus; diagnosis of tinnitus; special types of tinnitus; therapeutic approaches; psychological and behavioural aspects) and tips on how to handle tinnitus distress. ST included 64 different artificial and naturalistic sounds with various state of the art modulation or filter techniques. Loudness and length of the sounds was adjustable by the patients. There are many different SC and ST approaches administered by clinicians. For clarity, we want to mention that our app-based approach did not follow the Tinnitus Retraining Therapy protocol.”

Line number 466 – 471:

“Any interpretation of our findings should keep in mind, that we investigated specific applications of CBT, HA, ST, and SC. Potential reasons for the low efficacy of ST and SC in the present trial might include its self-administration, the limited interaction with a clinical specialist and/or the absence of specific instructions (stimulus, loudness, duration etc.). Thus, our conclusions on ST and SC might not be directly applied to a traditional clinical setting, where patients are not necessarily followed-up.”

The THI was not designed to measure outcome and with its limited scale lacks sensitivity to small but potentially important changes or differences. Hence it requires very large changes to suggest clinically meaningful change. This should be listed as a limitation.

>RESPONSE: We are aware that the THI has not been primarily designed to measure outcome. Nonetheless, the THI has been widely used in the past as outcome measurement tool in clinical trials (Hall et al., 2016, DOI: 10.1186/s13063-016-1399-9) and has been shown to be sensitive for changes. A recent study has compared the THI, the TFI and the TQ for assessing treatment effects in over 200 tinnitus patients at baseline as well as post treatment and demonstrated high conformity (Boecking et al., 2021, DOI: 10.3389/fpsyg.2021.596037). Our choice of the THI as primary outcome measure was based on consensus recommendations and on the fact that the THI is the most widely used questionnaire in the in both research and clinical management. (Hall et al., 2016, DOI: 10.1186/s13063-016-1399-9; Fuller et al., 2017, DOI: 10.3389/fpsyg.2017.00206; Langguth et al., 2007, DOI: 10.1016/S0079-6123(07)66050-6). In order to compensate for the weaknesses of the THI, we used additional secondary outcome measures (including the TFI) – please see tables 2 & 3 as well as the Supplementary Material, where we provide material for interested readers. We added a justification statement why we used the THI as a primary outcome measure.

Line number 232 – 235:

“Despite some critique on its sensitivity,^{27,28} the THI was chosen as the primary outcome measure, since i) it is the most widely used clinical instrument in research,^{29,30} and ii) there is high evidence of a conformity between the THI, the Tinnitus Functional Index (TFI)³¹, and the Tinnitus Questionnaire (TQ).³²”

Lack of acknowledgement of the established Core Outcome Set for tinnitus intervention trials should also be addressed. See Hall et al. (2018) The COMiT’ID study: developing core outcome domains sets for clinical trials of sound-, psychology-, and pharmacology-based interventions for chronic subjective tinnitus in adults. Trends in hearing, 22, 2331216518814384.

>RESPONSE: Thank you very much for pointing this out. We added the following part to the Methods section about our outcome measures and cited the paper you mentioned.

Line number 244 – 249:

“There is expert-based consensus on which outcome domains should be ideally assessed in tinnitus trials. However, there is still no consensus-based recommendation on which standardised instruments should be used within the selected outcome domains.³⁷ Different secondary outcome measures were considered here to underpin interpretability, validity as well as comparability of potential findings with past and future research.”

Why use multiple measures of tinnitus outcome (TFI, mini-TQ, 6 numerical scales) as secondary measures? If to capture different tinnitus outcomes, please specify.

>RESPONSE: In our opinion, the usage of only one outcome measure would drastically restrict the interpretability of our findings, as it not only opens the question about the validity of a single instrument used to capture changes in certain dimensions etc., but also hampers comparability as various studies, clinics, and labs use different assessments. In a situation where the choice of the most appropriate outcome measurement to cover all relevant aspects

of tinnitus is still a matter of debate, it seemed most appropriate to use several outcome measurements, as this provides more comprehensive insights in the effects of the various treatments and contributes to further refinement of outcome measurement in the tinnitus field. We added a short justification statement to the Methods section on why we used several secondary outcome measures:

Line number 247 – 249:

“Different secondary outcome measures were considered here to underpin interpretability, validity as well as comparability of potential findings with past and future research.”

Missing data doesn't appear to be completely missing at random – hearing aid recipients appeared the least compliant. Any reason for this?

>RESPONSE: We infer that you are referring to Figure S1 in the Supplementary Material.

From the figure it becomes evident, that besides HA, also SC and ST showed a similar level of non-adherence. We can only speculate about the reasons for this. We updated the relevant section in the discussion:

Line number 477 – 497:

“Despite the usage of interventions allowing for a high level of patient flexibility (SC and ST via mobile applications, HA), treatment compliance/adherence was low (see **Figure S1** and **Table S35**) and dropout rates were high in our trial (**per-protocol (PP)** sample of 185 patients). **CBT treatment arms, which require a high level of commitment with several on-site visits, demonstrated the highest proportion of dropouts in our trial, which potentially limits the interpretability and robustness of our CBT findings, as non-responders may be overrepresented among dropouts. In another recent study, in which CBT was compared with Neurofeedback, the CBT dropout rate was in a similar high range like in our study.⁵³ There is a large body of evidence in the literature that CBT is effective in the treatment of tinnitus (for an overview see the Cochrane review by Fuller et al., 2020),³⁹ and has been recommended in European**

guidelines for the management of tinnitus.⁸ However, all studies investigating CBT alone might be susceptible to a selection bias, as only patients with motivation for CBT would have been enrolled. The relatively high dropout rate of CBT in studies comparing various treatment options reflects the clinical experience of the real-world situation where a relevant subgroup of patients is not willing to undergo CBT. Detailed information on dropout reasons per treatment arm are listed in **Tables S6 – S9**.

With the application of two treatments in combination, the chances that one or even both treatments are not conducted as intended are increasing. **The lack of monitoring, strict guidance, or outpatient care in the case of SC, ST and HA, might be further potential reasons for treatment non-adherence.** Furthermore, high dropout rates are a well-known issue in mobile health interventions.⁵⁴ Another reason could be that patients were randomized to treatments and did not receive the treatment they desired. Under ideal treatment compliance/adherence (PP analysis), we observed no overall superiority of combination treatments.”

Discussion should acknowledge the added benefit of hearing aids for hearing loss, and the limitation that this could be conflated with tinnitus benefit.

>RESPONSE: Thank you for this point. We updated the discussion section and also added the information, that the recommendation for HAs is restricted to the treatment of hearing loss plus also that the benefit for hearing loss could be conflated with an amelioration in tinnitus symptoms:

Line number 446 – 460:

“This is the first systematic trial to investigate CBT, HA, ST, and SC within the scope of one investigation. **CBT approaches demonstrate the best body of evidence so far and are thus recommended by current treatment guidelines.^{7,8,47} Of today, the recommendation for HAs is restricted to the treatment of concomitant hearing loss, and there is no recommendation for ST due to a lack of clear scientific evidence.^{48–50} Counselling is recommended in form of**

information about tinnitus and the learning of potential coping strategies. However, counselling is usually not systematically structured and not investigated as such.⁵¹

With the present trial, we can directly put into perspective the effect size of CBT as the most established **evidence-based** treatment in tinnitus,^{7,8,39,47,52} with HA, ST, and SC (ST and SC provided with mobile applications) as well as their combinations as treatment options for tinnitus. Further, the present trial provides the first large-scale evidence for HA and SC (administered as stand-alone treatments), with a clinical effectiveness on a similar level as CBT. In view of the interpretation of the present findings for HAs, it is important to point out that the primary focus of a HA is on reducing hearing impairment by amplification of peripheral sounds and that this benefit could be conflated with an amelioration in tinnitus-related symptoms.”

Reviewer #4 (Remarks to the Author):

The manuscript titled "Single versus Combination Treatment in Tinnitus: An International, Multicentre, Parallel-arm, Superiority, Randomised Controlled Trial" presents the results of a multinational research project aimed at assessing clinical effectiveness of a range of single treatments vs. selected combination treatments in the treatment of patients with tinnitus.

The study enrolled and randomized 461 patients with chronic tinnitus and mild or more severe tinnitus handicap across 5 sites (2x Germany, Belgium, Greece, Spain). Treatment groups consisted of single treatments with CBT, HA, SC, or ST, as well as their pairwise combinations. The primary outcome was the difference in change in THI from baseline to 12 weeks in single vs. combination treatments.

General Remarks:

The manuscript is well written and summarizes a complex research endeavor concisely, while at the same time providing a large range of information on the study conduct and proceedings as well as more detailed and sensitivity analyses results in appendices.

>RESPONSE: We thank the reviewer for the positive feedback and the careful review. Below we provide answers to his/her comments, including helpful points in the manuscript.

However, the research question, choice of study design, and lack of country-specific sensitivity analyses raise questions which should be addressed to provide a fully nuanced account of the study. Firstly, the primary outcome strikes me as odd. Comparing (the mean of) all individual interventions with (the mean of) all combination treatments would suggest that the authors believe there is sufficient homogeneity in single and combination treatments per se as to account for a possible effect. In the discussion, the authors acknowledge that CBT has a record of effectiveness in tinnitus therapy, while ST performed poorly. If the authors believed there was heterogeneity in clinical effectiveness of

single interventions, the more natural research question in my mind would be if each individual single therapy option benefits from a combination with another method or not. On the other hand, the current research question would yield valid insights into the real-world effectiveness of treatment on a population level if the distribution to single treatments were representative of that in actual patient care. However, as treatment groups were approximately balanced, it remains unclear what to learn from the aggregate effectiveness of single treatments. While I do not suggest that the authors change their original research question, it would help to understand the research question better if the apparent inconsistencies are addressed. In addition, presenting EMMs over time for single interventions in one graph would help to understand the variation in treatment effect by intervention.

>RESPONSE: We thank the reviewer for the valuable comment. The primary aim of our study was to assess whether combination therapies generally exhibit superiority over single treatments, overcoming the variability inherent in individual therapies. Additionally, we aimed to investigate whether each single therapy benefits from combination with another treatment (as illustrated in Figure 2C-H), which aligns with your suggestion. Thus, our analysis focused on both aspects, whether combination therapies were universally more beneficial than single therapies as well as a more detailed examination to identify which specific combinations are most effective. We have opted for a controlled, balanced design to be able to attribute effects to the treatments themselves. Please see the Supplementary Material for a longitudinal effect of each treatment arm (Table S6).

We hope that this addresses the inconsistencies that you are referring to.

Secondly, a multi-national study design was chosen, with sites across 4 countries in the EU. While the multi-national setup might be warranted by the funding grant for this research, I feel the current manuscript misses the opportunity to address potential variations in

*treatments and their effectiveness across the countries involved. While I acknowledge that a full investigation into location-specific effects might shift the focus of the manuscript and exceed capacity, I also believe the current results must be viewed in light of geographic location and differences in healthcare services. I would suggest to include a sensitivity analysis with "country * objective" as additional fixed effect and reporting EMMs by country.*

>RESPONSE: Thank you for raising this concern. We fully acknowledge that differences in healthcare services may exist between countries. However, it is important to emphasize that this study was conducted outside the typical healthcare setting of the respective country. To ensure consistency across the study, workshops were held to standardize the study procedures. Two of the four treatment strategies were unguided, app-based therapies, minimizing potential differences. Additionally, the fitting of Hearing Aids (HA) was standardized, and the Cognitive Behavioral Therapy (CBT) strategy was co-developed specifically for this study – we also added this part to manuscript. Currently this information was only available from the study protocol.

Line number 191-197:

„ All treatment procedures were designed by dedicated experts in their respective fields (see **Table S3** for expert team per treatment type) and described in detail in the study protocol.¹⁷ To ensure consistency with respect to treatment and assessment implementation across clinical sites, workshops were held, and Standard Operation Procedure documents were created. Two of the four treatment types were unguided, app-based therapies, minimising potential differences. HA fitting was standardised, and CBT was co-developed specifically for this trial.“

While we agree that center-specific differences may still exist, this was minimized by harmonizing baseline parameters. We accounted for the variability induced by the different centers in our model. An analysis of center effects within the 10 treatment arms would not be very meaningful in our sample in view of the resulting small number of participants per cell.

Also, the authors report the trial has been impacted by the COVID-19 pandemic and was terminated prematurely (but without a practical loss in statistical power). As countries were affected differently across Europe, it would help to see the group allocation not only broken down by treatment group, but also by country to understand if recruitment and/or dropouts were affected differently by country.

>RESPONSE: Indeed, each clinical site had to adhere to certain local requirements due to COVID-19. We added this part to the manuscript:

Line number 313-315:

“..., plus the trial was performed during the COVID-19 pandemic with country-specific hospital policies, recruitment and inclusion processes lasted longer than expected.”

The direct impact of COVID infections was negligible: we only had 8 COVID-19 infections during the treatment phase (see Table S11) and only one dropout during the treatment phase with COVID-19 as self-reported reason (see Table S8). Splitting this low number up per country would not provide additional information.

According to your suggestion we added the total number of recruited patients per site to the manuscript:

Line number 317:

(Granada: 89, Athens: 99, Leuven: 74, Regensburg: 100, Berlin: 99).”.

Thirdly, approximately. 20% of MFI data was reported as missing. The authors assume data was MAR, and describe the procedures for assessing missing data in the SAP. While an assessment of the missingness mechanism always remains speculative, and some drop-out data hints at treatments being perceived as either not effective or impractical, it would help to put results into context if the authors report results under MNAR conditions.

>RESPONSE: We appreciate the reviewer’s insightful comment regarding the missingness

mechanism. As highlighted, the missingness mechanism cannot be definitively proven. However, our multiple imputation model incorporated relevant covariates and modeled the hierarchical structure of the data using multiple imputation by chained equations. This approach is considered the gold standard for dealing with missing data in epidemiological research and biostatistics (Van Ginkel et al., 2020; DOI: 10.1080/00223891.2018.1530680). While the MAR assumption is inherently unprovable, incorporating more covariates into the imputation model increases the plausibility of the MAR assumption, as it accounts for relationships that might explain the missingness mechanism (Van Ginkel et al., 2020, DOI: 10.1080/00223891.2018.1530680; Pedersen et al., 2017, DOI: 10.2147/CLEP.S129785; Grund et al., 2018, DOI: 10.1177/109442811770368). We updated the statistical analysis section in the manuscript accordingly:

Line number 269-272:

“For the ITT analysis, missing values (THI: 18%, education: 3.5%, PHQ-9 baseline: 2.6%) were imputed using multilevel imputation (R package mitml)^{40,41}; see **Figure S2** for the distribution of imputed THI values. **This approach is considered the gold standard for dealing with missing data.**⁴²”

To conclude, the current manuscript provides an excellent basis for expanding results and making the assumptions and conclusions more traceable.

>RESPONSE: Thank you very much for your positive feedback.

Response to the reviewers of the manuscript entitled:
**Single versus Combination Treatment in Tinnitus: An International, Multicentre,
Parallel-arm, Superiority, Randomised Controlled Trial**

Regensburg, May 4, 2025

Dear reviewers,

thank you very much for the positive and constructive feedback on the revised version of our manuscript as well as for appreciating our work. Please find our answers in a point-by-point format below each of your comments and raised points. Changes in the revised version of the manuscript are highlighted in red and appended to each response together with the line number. We further updated the order of tables and figures in the Supplementary Material to ensure consistency with their appearance in the main text.

Hopefully, the changes to our manuscript are appropriate and you will consider the revised version for publication.

Yours sincerely,

Stefan Schoisswohl
(on behalf of all authors)

Reviewer #1 (Remarks to the Author):

The authors have sufficiently addressed my concerns; exciting and impactful work!

>RESPONSE: Thank you very much. We are pleased to hear that you find our work exciting.

Reviewer #2 (Remarks to the Author):

157 and 166

Please state how subjects were asked to score the THI.

>RESPONSE: We added more information about the THI and how the 25 items of the THI were scored:

Line 229 - 235:

“The primary outcome between single and combination treatment was the difference in **total score** change from baseline to final visit (after 12 weeks of treatment) in the Tinnitus Handicap Inventory (THI).²⁰ The THI consists of **25 items designed to evaluate the perceived impact of tinnitus on an individual’s daily life. Each item provides three response options: “No”, “Sometimes” and “Yes”, which are scored as 0,2 and 4 points respectively. The total THI score is obtained by summing the scores of all items, resulting in a score that ranges from 0 to 100, with higher scores indicating greater perceived handicap due to tinnitus.**”

Widely used does not mean valid.

>RESPONSE: Thank you very much for this important information. Indeed, *widely used* does not mean valid. Since the THI was the only questionnaire available and validated in all necessary languages (Greek, German, Spanish, Dutch) at the time of planning and trial registration, we added this important information to our manuscript.

Line 236 - 242:

“Despite some critique on its sensitivity,^{27,28} the THI was chosen as the primary outcome measure, since i) it is the most widely used instrument in **clinical settings and is recommended as an outcome for clinical trials based on expert consensus,**²⁹⁻³¹ ii) there is high evidence of a conformity between the THI, the Tinnitus Functional Index (TFI)³², and the Tinnitus Questionnaire (TQ),³³ **plus iii) a validated version was available in the required languages (Dutch, German, Greek, Spanish) at the time of trial registration and the definition of our primary outcome measure.**^{34-37”}

The THI actually uses a 3-label category scale. Your numeric total scores, suggest you did not actually use the THI.

>RESPONSE: As already outlined above, we added the relevant information on how our patients scored the THI. We hope this information is sufficient.

Line 229 - 235:

“The primary outcome between single and combination treatment was the difference in **total score** change from baseline to final visit (after 12 weeks of treatment) in the Tinnitus Handicap Inventory (THI).²⁰ The THI consists of **25 items designed to evaluate the perceived impact of tinnitus on an individual’s daily life. Each item provides three response options: “No”, “Sometimes” and “Yes”, which are scored as 0,2 and 4 points respectively. The total THI score is obtained by summing the scores of all items, resulting in a score that ranges from 0 to 100, with higher sores indicating greater perceived handicap due to tinnitus.”**

Reviewer #3 (Remarks to the Author):

Thank you for the response to reviewer comments and revisions made. I do not have further to add but to reiterate,

>RESPONSE: Thank you very much for your positive feedback with respect to our revised manuscript. We tried to answer your raised points below.

The trial is not truly random but I appreciate a stratified randomization procedure was used.

>RESPONSE: We are glad that you agree with the updated section about the randomization approach of our trial.

I disagree with the authors use of the term Counselling in place of self-help/self-management.

>RESPONSE: We appreciate your feedback, and we can understand your criticism. Although there is no generally accepted clear definition of counselling, many people may have the idea that counselling implies face-to-face interaction with a professional or therapist. However, as already outlined in our previous response, there is growing evidence in favor of digitally delivered counselling content.

The use of digitally delivered counseling in our study was motivated by the desire to achieve the highest possible degree of standardization. The SC protocol was developed in accordance with the European Tinnitus Guidelines by an international panel of experts in the field of tinnitus and psychological interventions. The panel and also the UNITI consortium agreed on the labelling “Structured Counselling”, which was then used in the registration of the trial and also used in our published study protocol and statistical analysis plan.

Considering all these aspects, we consider a change of the term “structured counseling” at this stage as difficult and hope you understand our decision. However, we agree that it is mandatory to be entirely transparent about the type of counseling delivery and therefore we clearly state in our manuscript that our counselling approach is app-based and also address this issue specifically in our discussion (e.g., lines 484-489: *“Any interpretation of our findings should keep in mind, that we investigated specific applications of CBT, HA, ST, and SC. Potential reasons for the low efficacy of ST and SC in the present trial might include its self-administration, the limited interaction with a clinical specialist and/or the absence of specific instructions (stimulus, loudness, duration etc.). Thus, our conclusions on ST and SC might not be directly applied to a traditional clinical setting... “*).

I consider the report incomplete without the 48-week data. Will this data be made available, e.g. for critique in systematic reviews – it is not clear from the data availability statement. Also would rephrase your rationale for not reporting it - all research data is collected from volunteers.

>RESPONSE: We want to apologize for our misleading wording concerning the 48-week data and want to clarify this issue. Currently there is no effort from our side to analyse the 48-week follow-up. Reason for this is that the 48-week follow-up was on a voluntary basis and, consequently there are a lot of missing data so that no reliable conclusions can be from such analysis. However, we confirm that all data, including the 48-week follow-up data, will be made publicly available.

Therefore, we rephrased this section as such.

Line 222 – 226:

“An additional follow-up visit (~~48 weeks after baseline~~) was conducted **48 weeks after baseline**. **This visit was a voluntary follow-up visit. Due to a large amount of missing data (only 32.54% of participating patients provided data), no reliable conclusions can be drawn from the analysis and therefore** this additional follow-up was not included in the final outcome measure analysis.”

We are currently preparing a dataset description manuscript and are planning to upload the data to an online repository after publication of the manuscript at hand. For clarification, we updated our data availability statement as follows:

Line 634 - 643:

“De-identified data **reported and analysed here will be available upon request to the corresponding author. The complete dataset (incl. 48-week follow-up) and its description is currently under preparation for publication and release** via ZENODO. Status of the data availability will be **updated** on the UNITI website (<https://uniti.tinnitusresearch.net/>).”

I am unconvinced on the choice of THI as the primary outcome. Your strongest argument is that it conforms with the TFI and TQ – so why not use the TFI then which by design has greater sensitivity to smaller changes with its 11-point item scales versus the 3-point scales on the THI.

>RESPONSE: We understand the reviewer's concerns regarding the THI. The decision on selecting the THI as primary outcome was done at the design stage of the present trial, on a consensus decision by all leading PIs at the trial institutions. This decision was subsequently registered and published in our study protocol and statistical analysis plan as the definition of a primary outcome prior trial start is essential for ensuring scientific rigor, transparency, and credibility. Among the reasons for selecting the THI is its wide use in clinical settings in Europe. Since we are not testing any new intervention, but existing ones and their combination thereof, we estimated that a metric consistent with current clinical assessments in EU was best.

In addition, the THI is the most widely used tinnitus questionnaire (<https://pubmed.ncbi.nlm.nih.gov/27250987/>) and has been recommended by a consensus paper (<https://pubmed.ncbi.nlm.nih.gov/17956816/>). We were well aware of the weaknesses of the THI and, and therefore we decided to use additional widely used and well-investigated questionnaires as secondary outcomes, in order to address potential concerns and ensure maximal comparability with other studies.

Finally, another reason why we used the THI as our primary outcome was based on the fact that at the time of trial registration and the definition of our primary outcome the THI was the only questionnaire of which a translated and validated version was available in all necessary languages. We added this information to the outcome measure section of our manuscript.

We hope the reviewer understands that at this stage we have to comply with our protocol and cannot change the primary outcome, since switching from the THI to the TFI would compromise the scientific integrity of our trial.

Lines 236 - 242:

“Despite some critique on its sensitivity,^{27,28} the THI was chosen as the primary outcome measure, since i) it is the most widely used instrument in **clinical settings and is recommended as an outcome for clinical trials based on expert consensus,**²⁹⁻³¹ ii) there is high evidence of a

conformity between the THI, the Tinnitus Functional Index (TFI)³², and the Tinnitus Questionnaire (TQ),³³ plus iii) a validated version was available in the required languages (Dutch, German, Greek, Spanish) at the time of trial registration and the definition of our primary outcome measure.³⁴⁻³⁷”.

Reviewer #4 (Remarks to the Author):

Thank you for the additional information provided in response to my comments and questions. However, I would still recommend to implement the following analyses to challenge your assumptions and strengthen the conclusions you draw from the results.

1) Although protocols for treatments have been harmonized across countries, differences in individual interventions' effectiveness might still exist due to variations in implementation and acceptance of the patient population for the individual treatments. A country-specific analysis of possible variations in treatment and their effectiveness is the prerequisite before drawing conclusions from the pooled data. I strongly recommend conducting this sensitivity analysis.

>RESPONSE: Thank you for your recommendation. We calculated least-square mean changes for the THI change from baseline to final visit for each treatment arm and each country (**Table S19** below). Pairwise post-hoc contrasts for the THI least-squares mean change revealed statistically significant (Bonferroni adjusted) differences between CBT + SC and ST, CBT + ST and ST only for Germany. Please take into account that there were two centers in Germany and only one center in the other countries resulting in different sample sizes of the country-specific analyses. We further added the following to our methods and results part.

Lines 296 - 299:

“The results of the remaining objectives as described in the SAP are reported in the Supplementary Appendix. **Additionally, we evaluated THI score changes from baseline to final visit for single and combination treatment as well as for all individual treatment arms separately by country to assess potential country-specific effects.**”

Lines 362 - 365:

The results of the remaining objectives (as outlined in the SAP)¹⁸ and time points (interim visit and follow up) are reported in **Tables S16 – S18**. **Country-specific changes for the THI from baseline to final visit for single and combination treatment as well as for all treatment arms can be found in Table S19.**

Table S19: THI Change from Baseline to Final Visit for each Country

Contrast	Belgium	Germany	Greece	Spain
Single	-9.4 [-3.2; -15.5]	-12.0 [-8.0; -16.0]	-16.0 [-9.2; -22.8]	-8.8 [-2.4; -15.2]
Combination	-13.1 [-4.9; -21.2]	-16.5 [-12.6; -20.4]	-15.8 [-9.3; -22.3]	-11.3 [-5.1; -17.5]
CBT	-19.1 [-5.7; -32.4]	-16.4 [-8.0; -24.8]	-20.4 [-6.8; -34.0]	-11.4 [2.6; -25.5]
CBT + HA	-38.0 [0.7; -76.7]	-14.1 [0.8; -29.1]	-11.1 [14.8; -37.0]	-16.0 [5.7; -37.6]
CBT + SC	-13.7 [6.5; -33.9]	-21.1 [-12.8; -29.4]	-15.7 [-2.0; -29.5]	-12.2 [2.8; -27.3]
CBT + ST	-12.9 [5.8; -31.6]	-17.5 [-10.0; -25.0]	-11.8 [2.2; -25.7]	-10.1 [2.3; -22.6]
HA	-12.6 [-1.1; -24.2]	-15.5 [-8.2; -22.8]	-21.2 [-7.5; -34.8]	-7.8 [5.0; -20.6]
HA + SC	-26.7 [-2.0; -51.5]	-19.5 [-6.8; -32.2]	-21.1 [2.3; -44.6]	-15.0 [4.6; -34.6]
HA + ST	4.7 [27.0; -17.7]	-9.9 [1.0; -20.9]	-30.3 [-12.6; -48.1]	-10.3 [5.6; -26.3]
SC	-7.1 [5.3; -19.5]	-12.7 [-4.8; -20.6]	-13.5 [0.3; -27.3]	-14.9 [-1.2; -28.5]
SC + ST	-11.5 [1.1; -24.1]	-14.2 [-6.9; -21.5]	-13.0 [-0.6; -25.5]	-9.3 [3.5; -22.0]
ST	-0.3 [11.5; -12.1]	-3.4 [4.0; -10.9]	-9.3 [4.1; -22.6]	-2.6 [10.4; -15.6]

Note. Values depict least-squares mean changes with 95% CI in square brackets. Comparison of single and combination treatments as well as all treatments regarding the THI difference from baseline to final visit for each country. Note that there were two centers in Germany (Berlin, Regensburg).

2) Although MICE is a widely adopted imputation approach for data believed to be MAR, the reasons for drop-out stated in tables S7 and S8 suggest that in many cases, drop-outs occurred due to dissatisfaction with an intervention or no perceived benefit and worsening tinnitus symptoms. This suggest that some portion of data is MNAR, which needs to be addressed by an appropriate imputation mechanism. I strongly suggest the authors conduct a sensitivity analysis using a reference-based imputation approach (e.g. <https://doi.org/10.1177/1536867X1601600211>) to challenge their assumptions.

Thank you again for this interesting work and your effort in addressing all comments and questions this far!

>RESPONSE: We would like to thank the reviewer for the constructive suggestion. As suggested, we performed additional robustness checks by conducting reference-based imputations (jump to reference, copy increments in reference, copy reference) as well as last observation carried forward and repeated the analysis of our primary outcome (single vs. combined treatments at final visit). The results are listed in **Table S36** (see below). These analyses revealed that our main result cannot be sustained under the assumption of MNAR. We updated our methods and results part accordingly and added a statement to our discussion.

Lines 281 – 284:

“Additional sensitivity analyses were performed in the primary outcome without imputation, three different reference-based imputation approaches (*jump to reference, copy increments in reference, copy reference, R package RefBasedMI*)^{48,49} assuming data is not missing at random and the method of Last Observation Carried Forward

Lines 388-391:

“Sensitivity analyses of our primary outcome using no imputation and the method of Last Observation Carried Forward yielded similar results as our ITT analysis. However, under the assumption that data is not missing at random, our ITT findings cannot be upheld (**Table S35 – S36**).”

Lines 534-539:

“Even though in 18% of all participants data of the primary outcome (**THI**) was missing, the sensitivity analysis using no imputation came to similar findings, which was further corroborated by applying the Last Observation Carried Forward approach. Yet, under the assumption of “missing not at random” and after conducting additional robustness evaluations

using three different reference-based imputation methods, our findings cannot be sustained (see **Tables S35-S36**).”

Table S36: Sensitivity Analysis – Robustness Check of the Primary Outcome (THI) at Final Visit using different Imputation Methods

Imputation method	Least-square mean change – single	Least-square mean change – combination	β estimate	p value
Multilevel imputation	-11.7 [-14.4; -9.0]	-14.9 [-17.7; -12.1]	3.2 [0.2; 6.1]	0.034
No imputation	-11.5 [-14.3; -8.7]	-15.1 [-18.0; -12.2]	3.6 [0.5; 6.6]	0.022
Reference-based imputation (J2R)	-12.3 [-15.2; -9.4]	-14.5 [-17.1; -12.0]	2.3 [-1.7; 6.2]	0.252
Reference-based imputation (CIR)	-12.3 [-15.2; -9.4]	-14.5 [-17.2; -11.7]	2.2 [-2.0; 6.4]	0.292
Reference-based imputation (CR)	-12.3 [-15.2; -9.4]	-14.5 [-17.2; -11.8]	2.2 [-1.9; 6.3]	0.281
LOCF	-9.5 [-12.0; -7.1]	-12.6 [-15.1; -10.2]	3.1 [0.4; 5.7]	0.024

Note. Results of the primary objective (THI; single vs. combination treatments) under different assumptions of the missing data mechanism. Depicted are least-square mean changes with 95% CI in square brackets and the results (β estimate with 95% CI in square brackets and p value) of the interaction effect (single vs. combination treatments at final visit vs. baseline). Reference-based imputation methods fill missing values following the distribution of a designated reference arm. We used the single treatment arm as reference (R package `RefBasedMI`). *Jump to reference (J2R)*: As soon as a participant has a missing value, all future outcomes are set to match the reference arm’s trajectory from that point onward. *Copy increments in reference (CIR)*: Missing values are imputed by preserving the participant’s last observed offset from the reference group but then applying the reference group’s subsequent changes. *Copy reference (CR)*: Missing values are imputed as if the participant were always in the reference group, effectively discarding any prior difference from the reference. *Last observation carried forward (LOCF)* replaces each participant’s missing values with their most recent observed measurement, assuming that the outcome remains unchanged after the last observation (single imputation method).

Response to the reviewers of the manuscript entitled:
**Single versus Combination Treatment in Tinnitus: An International, Multicentre,
Parallel-arm, Superiority, Randomised Controlled Trial**

Regensburg, June 16, 2025

Dear reviewers,

thank you very much for the positive and constructive feedback on the revised version of our manuscript as well as for appreciating our work. Please find our answers in a point-by-point format below each of your comments and raised points. We thank you for your time and effort to review our manuscript and give us the opportunity to improve our manuscript based on your suggestions and comments.

Yours sincerely,

Stefan Schoisswohl
(on behalf of all authors)

Reviewer #3 (Remarks to the Author):

Thank you for your further responses. I am still concerned that much of the study relied on pragmatic decision making and opinion, e.g., use of THI as a primary outcome measure.

I absolutely disagree there are no generally agreed definitions of counselling. For such you should look to the relevant societies rather than falling on non-expert perspectives, e.g. the American Counseling Association clearly defines counselling as "a learning-oriented process, carried on in a simple, one-to-one social environment, in which a counselor, professionally competent in relevant psychological skills and knowledge, seeks to assist the client, by methods appropriate to the latter's needs and within the context of the total personnel program, to learn more about himself and to accept himself, to learn how to put such understanding into effect in relation to more clearly perceived, realistically defined goals to the end that the client may become a happier and more productive member of his society."

British Psychological Society definition: Counselling and psychotherapy are forms of 'talking therapy' for personal issues, such as stress, worry, anxiety or depression. Counselling may be briefer and provides a listening space, while psychotherapy may be longer and look more closely at emotional or behavioural difficulties. They involve sharing your problems with your therapist or counsellor in a confidential setting.

A textbook definition of counselling is a contracted meeting between a client and a counsellor. The meeting happens at a set time, in an agreed place, for the sole benefit of the client.

Counselling happens at a specified time and at a specific place, and the sole focus of the meeting is to benefit the client.

>RESPONSE: We sincerely appreciate your thoughtful feedback and your time-effort to review our manuscript. We understand your concerns regarding the terminology of counselling. However, we would like to respectfully clarify that the term counselling is used differently in the specific context of tinnitus. In the field of audiology and tinnitus research/ treatment, counselling typically refers to psychoeducation and structured information delivery about tinnitus mechanisms, coping strategies as well as treatment options (please see the section procedures in our manuscript and our study protocol). This differs from counselling psychology

as defined by the American Counseling Association or the British Psychological Society. Tinnitus counselling is well established in the literature and clinical guidelines. It can include brief, structured sessions that are not equivalent to formal psychotherapy. It is not intended to replace professional mental health interventions, but rather to support patients through education and expectation management, often delivered by audiologists or trained clinicians rather than licensed psychotherapists. We hope we could clarify this terminological misunderstanding and kindly ask for your understanding to keep the term counselling. Moreover, the term has been used consistently in our study-related materials (study protocol, statistical analysis plan, study registration etc.).

With respect to our primary outcome measure, we hope you understand that at some point during the study planning phase the decision has to be made on which primary outcome shall be used. Our decision to use the THI has been thoroughly explained in our previous responses and is e.g., based on widespread usage, language availability and validation. We hope you can follow our reasonings.

Reviewer #4 (Remarks to the Author):

Thank you for including the additional analyses and discussing results more broadly! I have no further requests and thank the authors for their contribution.

>RESPONSE: We sincerely thank the reviewer for the positive feedback and constructive input throughout the review process. We appreciate your engagement and time-effort and are glad that the additional analyses and broader discussion addressed your concerns.